# Design principles for enhancing phase sensitivity and suppressing phase fluctuations simultaneously in biochemical oscillatory systems

Chenyi Fei [1], Yuansheng Cao[2], Qi Ouyang[1] & Yuhai Tu[3]

Biological systems need to function accurately in the presence of strong noise and at the same time respond sensitively to subtle external cues. Here we study design principles in biochemical oscillatory circuits to achieve these two seemingly incompatible goals. We show that energy dissipation can enhance phase sensitivity linearly by driving the phase-amplitude coupling and increase timing accuracy by suppressing phase diffusion. Two general design principles in the key underlying reaction loop formed by two antiparallel pathways are found to optimize oscillation performance with a given energy budget: balancing the forward-to-backward flux ratio between the two pathways to reduce phase diffusion and maximizing the net flux of the phase-advancing pathway relative to that of the phase-retreating pathway to enhance phase sensitivity. Experimental evidences consistent with these design principles are found in the circadian clock of cyanobacteria. Future experiments to test the predicted dependence of phase sensitivity on energy dissipation are proposed.

[1] The State Key Laboratory for Artificial Microstructures and Mesoscopic Physics, School of Physics, Peking University, Beijing 100871, China. [2] Department of Physics, UC San Diego, La Jolla, CA 92093, USA. [3] IBM T. J. Watson Research Center, Yorktown Heights, New York, NY 10598 USA. These authors contributed equally: Chenyi Fei, Yuansheng Cao. Correspondence and requests for materials should be addressed to Y.T. (email: yuhai@us.ibm.com)

Biochemical processes are innevitably noisy due to the stochastic nature of reactions, the small number of molecules involved, and the thermal fluctuations from environment[1,2]. Various regulatory mechanisms have evolved to suppress effects of noise in order to process information accurately in vital life processes such as biomolecule synthesis[3], cell cyle[4], and development[5]. At the same time, some of these systems also need to have a high sensitivity to external stimuli. For example, for many biochemical oscillatory systems, such as glycolysis, cyclic AMP signaling, cell cycle, circadian rhythms, and neural activities[4,6–8], besides being accurate in their rhythmic timing, they also need to respond sensitively to external cues. In fact, one of the most salient properties of circadian rhythms is their ability to be entrained by the daily cycle in the environment so that their endogenous 24 h cycle can quickly synchronize with environmental signals[9,10].

However, these two requirements, high sensitivity and low fluctuation, are incompatible for equilibrium systems due to the Fluctuation Dissipation Theorem (FDT)[11]. Briefly, for a perturbation of intensity $\epsilon$ applied to the conjugate variable of an observable $A$ at time $t = 0$, FDT establishes the fluctuation-response relation (FRR) $\langle A(t)\rangle_\epsilon - \langle A\rangle_0 = \beta\epsilon[C_A(t, t) - C_A(t, 0)]$, where $C_A(t, s) = \langle A(t)A(s)\rangle_0$ is the two-time autocorrelation and $\beta = 1/k_BT$ is the reverse thermal energy. We immediately see that the long time response $\Delta A \equiv \langle A(t = \infty)\rangle_\epsilon - \langle A\rangle_0$ is linearly proportional to the variance, i.e., $\Delta A = -\beta\epsilon\sigma_A^2$. This means that a higher sensitivity ($\Delta A/\epsilon$) would necessarily lead to a higher fluctuation ($\sigma_A^2$) in any equilibrium system. Such a FRR was also found in certain biochemical systems in their linear response regime[12].

To understand how living organisms solve the challenge of enhancing sensitivity (responsiveness) and reducing noise (fluctuation) at the same time, we studied the dynamics of a large class of biochemical oscillators in which limit cycles exist with the focus on non-equilibrium effects in the underlying biochemical reaction networks where FRR breaks down. Recently, the relationship between biological regulatory functions and their energy cost has attracted much attention in non-equilibrium statistical physics community[13–18]. A previous study found that the phase diffusion constant can be suppressed by a dissipative process that consumes free energy[19]. In this work, by studying different types of limit cycle oscillators analytically and numerically, we investigated whether dissipative processes can enhance sensitivity and

reduce fluctuation at the same time. More importantly, our study uncovered the key design principles for biochemical circuits to achieve these two goals simultaneously.

## Result

**Reduced phase description for biochemical oscillations.** The dynamics of a biochemical reaction system $\{X_1, X_2, …, X_N\}$, with fixed volume $V$ and constant temperature, is described by chemical Langevin equation (CLE)[20]. For a biological oscillator, the concentration variable of $i^{\text{th}}$ species $x_i(t)$ oscillates. Instead of dealing with the entire system, we employ the phase reduction method first developed by Kuramoto[21–23], which reduce $N$-dimensional state space to a single phase variable $\phi$ characterizing the timing of oscillation. Specifically, $\phi(x^{\mathcal{L}})$ along the deterministic limit cycle $\mathcal{L}$ is chosen to progress with a constant speed $\Omega = 2\pi/\tau$ for convenience, where $\tau$ is the period. This definition of the phase can be extended to the whole basin of attraction of $\mathcal{L}$[24]. If trajectories originated from two states eventually converge onto the limit cycle at the same time, these two states are assigned the same phase. An isochron is a line formed by all points with the same phase (see Fig. 1a).

Clearly, geometrical structure of isochrons is crucial to the phase response property: larger $\nabla_x\phi$ would produce larger phase shifts for the same deviation from limit cycle[25]. In biology literature, a phase response curve (PRC) is commonly used to characterize oscillators' responsiveness[26–29]. The PRC $\Delta\phi(\phi)$ is determined by delivering a perturbation at a given phase $\phi$ of the oscillation for a given duration of time and comparing the shift in peak times between the perturbed trajectories and the unperturbed ones to obtain $\Delta\phi$[30]. Indeed, as shown in Supplementary Note 1, $\nabla_x\phi$ is the key signal-independent factor in determining the amplitude of PRC. At a given phase, we define a dimensionless phase gradient vector $\nabla_{x^*}\phi$ where $x^* = x_i/(x_i^{\max} - x_i^{\min})$ is a dimensionless state variable (normalized by the range of variation in $x_i$). We further use the maximal Euclidean norm of $\nabla_{x^*}\phi$ along the limit cycle to define a global phase sensitivity parameter $\chi$:

$$\chi \equiv \max\{\|\nabla_{x^*}\phi\|\}. \qquad (1)$$

Throughout this work, we assume that change in other signal-dependent factors (see Supplementary Note 1) does not overwhelm the effect of $\chi$ on phase shift.

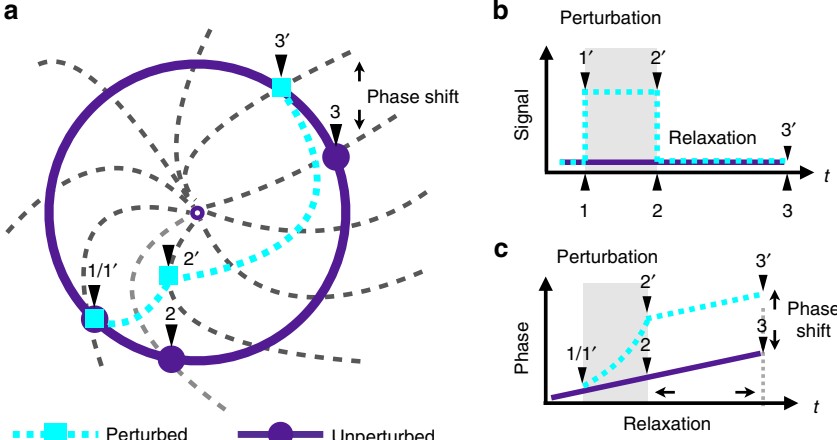

**Fig. 1** Illustration of the phase response in biological oscillators. **a** The circle is the assumed stable limit cycle. The gray dashed lines represent equally separated isochrons. An unperturbed system (purple) progresses on the circle, while a perturbed system (cyan) is driven away from the circle by an impulsive signal between time 1 and time 2, and then relaxes back to the limit cycle. At time 2 (end of perturbation), it is moved to an isochron different from the unperturbed one. The difference of their phases determines the phase shift. **b, c** Diagrams of the signal and phase evolution of the perturbed and unperturbed system. Phase shift is induced during the perturbation and sustains after the perturbation

Intuitively, a more sensitive circuit can enhance entrainment as it can more readily change its oscillation phase to sync with external stimuli[31]. It is also straightforward to show analytically that a higher sensitivity widens the range of synchronizable frequencies known as the Arnold tongue[32] and therefore enhances entrainability of biochemical oscillators (see Supplementary Note 2 for details).

Due to the stochastic nature (Poisson process) of the underlying chemical transitions, biochemical oscillations are noisy and the phase fluctuates[19]. The variance of phase fluctuations $\sigma_\phi^2$ grows linearly with time, $\sigma_\phi^2 = D_\phi t$, which can be used to define a phase diffusion constant $D_\phi$. It can be shown using the phase reduction method that the probability distribution of phase fluctuation ($\delta\phi$) indeed follows a diffusive dynamics $\partial_t P(\delta\phi, t) = D_\phi \partial_{\delta\phi}^2 P$ (see Supplementary Note 3). It is usually hard to directly derive the phase equation. As shown in Supplementary Note 4, the finite correlation time $\tau_c$ due to phase diffusion is inversely proportional to $D_\phi$. Thus, we use this relationship to infer the phase diffusion constant from the autocorrelation function, which follows a damped oscillation[19]

$$C(t, 0) = C_0 \exp(-t/\tau_c) \times \cos(\Omega t) \qquad (2)$$

We now introduce the specific biochemical oscillators we study here. It is known that nonlinear autocatalytic biochemical reactions in open systems can exhibit oscillatory behaviors. For clarity, we employ a simple model (Fig. 2a) derived from

glycolysis oscillation[33] and the Brusselator[34,35] as follows:

$$
\begin{array}{ccc}
k_1 & k_2^0 & k_3 \\
A \rightleftharpoons X; & B + X \rightleftharpoons D + Y; & 2X + Y \rightleftharpoons 3X \\
k_{-1} & k_{-2}^0 & k_{-3}
\end{array}
\qquad (3)
$$

where $A$, $B$, and $D$ have fixed concentrations. Here we neglect the inhomogeneous spatial distribution and focus on the dynamics and energetics of the "well-stirred" reaction system.

To study thermodynamics of the system properly, we include the backward reactions (see Fig. 2a) in the Brusselator model. An equilibrium steady state can be achieved when $k_{-2}^{(eq)} k_{-3} = k_2^{(eq)} k_3$, where $k_2^{(eq)} = k_2^0 [B]^{(eq)}$, $k_{-2}^{(eq)} = k_{-2}^0 [D]^{(eq)}$ are pseudo-first-order rate constants and equilibrium values are labeled by superscript (eq). However, if the concentrations of $B$ and $D$ are sustained at values different from their equilibrium values by active processes such as biochemical synthesis and/or active transport (pumping), the system is driven out of equilibrium with the chemical potential difference $\Delta\mu_{DB} = -k_B T \ln(k_{-2} k_{-3}/k_2 k_3)$ serving as the chemical driving force for the reaction cycle $X \to Y \to X$.

To characterize the nonequilibrium cycle dynamics, we introduce a (global) irreversibility parameter $\gamma$:

$$\gamma \equiv \frac{k_{-2} k_{-3}}{k_2 k_3} = \exp(-\Delta\mu_{DB}/k_B T), \qquad (4)$$

which is related to the chemical driving force for the reaction cycle (see Fig. 2). For the special case of an equilibrium system with $\gamma = 1$, detailed balance is satisfied and the cycle is fully reversible without net flux. When $\gamma \neq 1$, the system is driven out of equilibrium by external free-energy sources resulting in a nonzero net cycling flux.

In general, external free energy can be utilized to change the chemical reactions in two ways: enhancing forward reactions or suppressing backward reactions. To understand effects of these two distinct changes, we further introduce $\gamma_1$ and $\gamma_2$ to characterize the (local) irreversibility of the forward and backward reaction respectively, i.e., $k_2 = k_2^{(eq)}/\gamma_1$, $k_{-2} = k_{-2}^{(eq)} \times \gamma_2$, with $\gamma = \gamma_1 \gamma_2$.

In addition to the Brusselator model, we have studied another class of biochemical oscillators driven by the general activator–inhibitor (AI) mechanism (see Supplementary Fig. 1). In the AI model, oscillation is driven by the ATP hydrolysis energy and $\gamma^{-1}$ can be expressed as $\frac{[ATP][ADP]^{(eq)}[P_i]^{(eq)}}{[ATP]^{(eq)}[ADP][P_i]}$. The two "local" irreversibility parameters ($\gamma_1$ and $\gamma_2$) are also introduced in the AI model to characterize the non-equilibrium effects in different parts of the phosphorylation–dephosphorylation (PdP) cycle that dissipates energy to drive the oscillation. Dissipation outside of the PdP cycle is roughly independent of $\gamma$ and does not have a direct role in controlling the oscillation (see Supplementary Note 5A and Supplementary Fig. 2 for details).

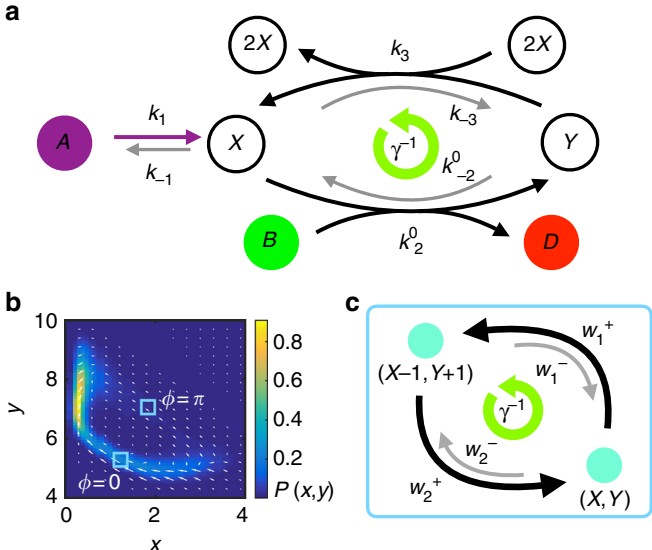

**Fig. 2** The reversible Brusselator model. **a** The reaction with $B$ and $D$ can be mapped into a unimolecular reaction with rate constants: $k_2 = k_2^0 [B]$, $k_{-2} = k_{-2}^0 [D]$. Together with the autocatalytic reaction with rates $k_3$ and $k_{-3}$, they form a reaction cycle with a reversibility parameter $\gamma \equiv k_{-2} k_{-3}/ k_2 k_3$. When $\gamma \neq 1$, the system is nonequilibrium with free-energy dissipation driving the cyclic flux $X \to Y \to X$. **b** The steady-state probability density $P(x, y)$ (color plot) and the state-space fluxes ($J_x$, $J_y$) (vector field) of the model. The two small blue boxes highlight the regions around the two opposite phases $\phi = 0$ and $\phi = \pi$ in the deterministic limit cycle. **c** Details of the chemical reactions in a local region (e.g., the small boxes in **b**) in the state space. Two microscopic states, $(X-1, Y+1)$ and $(X, Y)$, are linked by two distinct reversible reaction pathways, which form a microscopic reaction cycle (loop)

**Effects of free-energy dissipation on phase dynamics.** From the chemical reaction rates, we can compute the free-energy dissipation rate (in units of $k_B T$)[36]:

$$\dot{W} = \sum_i (J_i^+ - J_i^-) \ln \frac{J_i^+}{J_i^-} \qquad (5)$$

where $J_i^+$ and $J_i^-$ are the forward and backward fluxes of the $i^{th}$ reaction. As the dissipation rate also oscillates, we use $\Delta W = \int_0^\tau \dot{W} dt$ to measure free-energy cost per period per volume. Here we study how $\Delta W$ affects the performance of the oscillation as measured by its phase diffusivity and phase sensitivity.

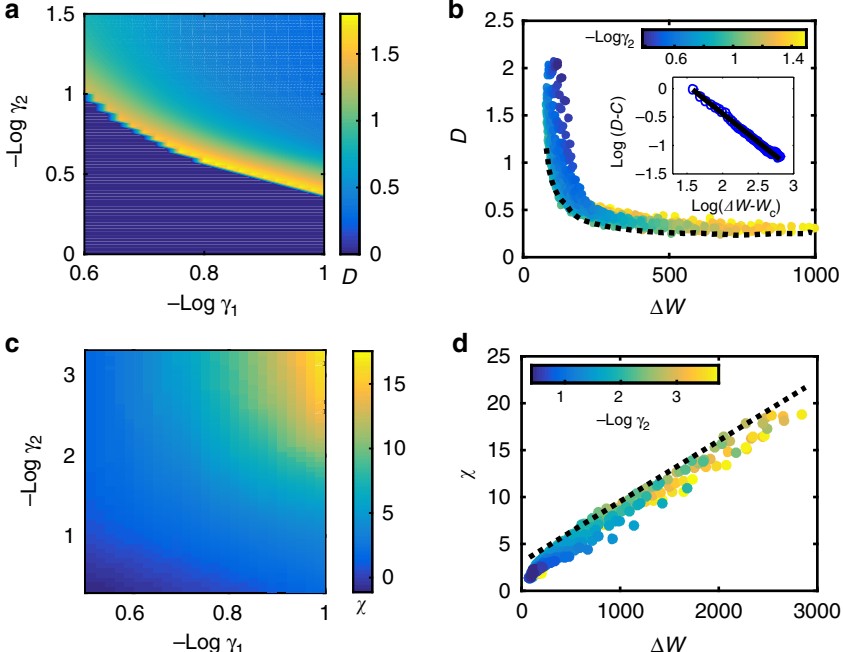

**Fig. 3** Energy dissipation suppresses phase diffusion and enhances phase sensitivity simultaneously. **a**, **c** Phase diffusion constant $D$ and phase sensitivity $\chi$ are plotted against non-equilibrium parameter $\gamma_1$ and $\gamma_2$. **b** $D$ depends inversely on energy dissipation per cycle $\Delta W$. (inset) The envelope curve (dashed line), which shows the minimum phase diffusion constant, is fitted well by $D_{\min} = C + W_0/(\Delta W - W_c)$[19]. The fitting parameters are: $W_c = 41.29$, $W_0 = 36.44$, and $C = 0.2305$. The volume of Gillespie simulation is $V = 100$. **d** $\chi$ exhibits linear dependence on $\Delta W$ and the maximum sensitivity $\chi_{\max} = K_W \Delta W$ + const. with the proportional constant $K_W \approx 6.5 \times 10^{-3}$

First, we briefly summarize the effect of energy dissipation on phase diffusion. As shown in Fig. 3a, b, the minimum phase diffusion constant $D_{\min}(\Delta W)$ depends inversely on the free-energy cost per period $\Delta W$ consistent with our previous work[19] and the general thermodynamic uncertainty relation for biomolecular processes[37]. Here we further dissect the separate dependence on $\gamma_1$ and $\gamma_2$. As shown in Fig. 3a, decreasing $\gamma_1$ or $\gamma_2$ can both reduce phase diffusion. However, $D$ eventually saturates to a nonzero value even in the absence of backward reactions ($\gamma_2 = 0$) due to the finite stochasticity in the forward reactions with a finitie $\gamma_1$ (see Supplementary Fig. 3b). On the other hand, $D$ seems to decrease continuously with $\gamma_1$ (see Supplementary Fig. 3c). As the number of forward reactions per unit time $N_c$ increases with $1/\gamma_1$, the averaging effect ($1/\sqrt{N_c}$ effect) in reducing the phase fluctuation can persist for small $\gamma_1$ (or large $N_c$). In fact, this noise reduction strategy, i.e., taking average over multiple irreversible steps, is also commonly employed in other biological processes[38].

We now turn to the main focus of this work, i.e., to understand how free-energy dissipation affects phase response. In particular, we ask whether low phase fluctuation and high phase sensitivity can coexist in a dissipative system. For different values of the forward and backward irreversible constants ($\gamma_1$ and $\gamma_2$), we calculated phase sensitivity $\chi$ of the biochemical oscillator to external perturbations (stimuli). Remarkably, we observed in Fig. 3c, d that the phase sensitivity $\chi$ is bounded by a maximum value $\chi_{\max}$ that increases linearly with $\Delta W$ in a wide range of $\Delta W$ for different combinations of $\gamma_1$ and $\gamma_2$:

$$\chi(\gamma_1, \gamma_2) \leq \chi_{\max}(\Delta W) = K_W \Delta W + \text{const.} \quad (6)$$

where $K_W$ is a constant whose value is given in the legend of Fig. 3. We have confirmed the generality of Eq. (6) for other implementations of $\gamma_1$, $\gamma_2$ (Supplementary Fig. 4) in the Brusselator model, as well as for the AI model (Supplementary Fig. 5).

**The relation between phase sensitivity and entrainment to external periodic driving.** Many biochemical oscillators are exposed to an external periodic signal $\epsilon p(\Omega, t)$ (e.g., temperature, light, etc.) that entrains the internal oscillation. The external signal varies with the system's intrinsic frequency $\Omega$ and acts on an internal parameter $\mu$. Incorporating this periodic driving force as time-dependent perturbation $f(t) = (\partial_\mu F)\epsilon p(\Omega, t)$ in the phase reduction description, we obtain the dynamical equation for the phase $\phi$

$$\frac{d\phi}{dt} = \Omega + Z_\mu(\phi) \cdot \epsilon p(\Omega, t), \quad (7)$$

where $Z_\mu(\phi)$ is the infinitesimal PRC function for perturbing $\mu$[25]. Applying a pulse perturbation with duration $\Delta t = \delta_t \tau$ and intensity $\Delta \mu = \delta_\mu$ on $\mu$, the PRC can be calculated as $\Delta\phi(\phi; \delta_t, \tau, \delta_\mu, \mu) = Z_\mu(\phi)\Delta t\Delta\mu$. We now relate the normalized PRC $C(\phi) = \Delta\phi(\phi)/(\delta_t\delta_\mu) = Z_\mu(\phi)\tau$ with the performance of entrainment to the external periodic driving signal.

To characterize the phase dynamics in the presence of an external periodic signal, we use the phase difference $\psi = \phi - \Omega t$ between the internal oscillator and the external signal. By averaging over a period (as $\psi$ is a slow variable when $\epsilon$ is small), we have

$$\frac{d\psi}{dt} = \frac{\epsilon}{\tau}\Gamma(\psi) \equiv \frac{\epsilon}{\tau} \times \frac{1}{2\pi}\int_0^{2\pi} C(\psi + \theta)p(\theta)d\theta. \quad (8)$$

Entrainment to the external signal occurs, because Eq. (8) has a stable fixed point $\psi_0$, i.e., $\Gamma(\psi_0) = 0$, $\Gamma'(\psi_0) < 0$. Now, we consider a sudden phase shift $\Delta\psi$ of the external signal, which is equivalent to an initial perturbation $\Delta\psi$ in $\psi$ away from its fixed point. When $\Delta\psi$ is small, the perturbation $\delta\psi = \psi - \psi_0$ follows, to the leading order, $\dot{\delta\psi} = -t_e^{-1}\delta\psi$ where $t_e = \left|\epsilon\Gamma'(\psi_0)\right|^{-1}\tau$ is the entrainment time for the system to catch up the shifted signal. It is clear that

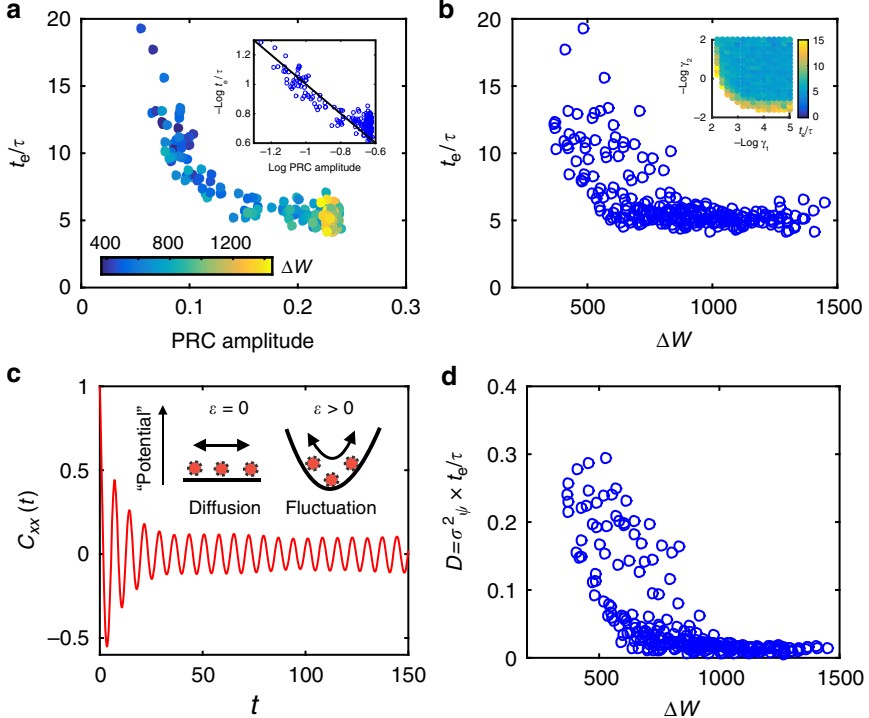

**Fig. 4** Relation between entrainment performance and phase sensitivity and free-energy dissipation. **a** The minimum number of cycles needed for entrainment $t_e/\tau$ versus the PRC amplitude for different choices of $(\gamma_1, \gamma_2)$. The energy dissipation per period $\Delta W$ is shown by the color of the data points. The log-log plot in the inset clearly shows $t_e/\tau$ scales inversely with the PRC amplitude. **b** $t_e/\tau$ decreases with $\Delta W$. The inset shows $t_e/\tau$ (represented by the color) in the $(-\log \gamma_1, -\log \gamma_2)$ space. **c** Autocorrelation function $C_{xx}(t)$ versus time $t$. The amplitude first decays exponentially and then reaches a constant $e^{-Dt_e}$ in agreement with Eq. (10). The phase dynamics in the cases without ($\epsilon = 0$) and with ($\epsilon > 0$) external driving signal are illustrated in the inset. **d** The phase diffusion constant $D$, inferred from phase variance, decreases with the energy dissipation $\Delta W$. The external driving signal strength $\epsilon = 0.1$ for all simulations

stronger signals (larger $\epsilon$) shorten the entrainment process. At the same time, the internal phase response property also has an important role. In general, both the steady state phase difference $\psi_0$ and the entrainment time $t_e$ depend on the shape of PRC ($C(\phi)$) and how the periodic signal is applied (i.e., the waveform $p(\Omega, t)$); however, as long as the shape of PRC remains approximately the same, enhancing phase sensitivity $\chi$ (the amplitude of PRC) reduces $t_e$:

$$t_e/\tau \sim \chi^{-1}. \tag{9}$$

We have simulated the AI model to test the above relation between the phase sensitivity $\chi$ and the entrainment time $t_e$ for a periodic driving $k_1^{(p)}(t) = k_1(1 + \epsilon \cos \Omega t)$. At different phases of the oscillation, a phase shift $\Delta \psi = 0.4\pi$ was applied to the external signal, triggering the entrainment process. The oscillator was considered "entrained" when the peak difference between a perturbed and unperturbed system is small $\delta\psi < 0.1\Delta\psi$. Minimum number of entrainment cycles $\min_\phi\{t_e\}/\tau$ was recorded for different $\gamma_1$ and $\gamma_2$. The results are shown in Fig. 4a, b for $\epsilon = 0.1$.

The external periodic signal also affects phase fluctuations, which is now bounded by a "potential" well established by the external signal (Fig. 4c). By approximating the potential well by its harmonic form, the stochastic phase evolution equation in the presence of noise (from stochastic chemical reactions) can be solved to determine the phase variance:

$$\sigma^2(t) \equiv \langle \delta\psi^2(t) \rangle = Dt_e\left(1 - e^{-t/t_e}\right)/\tau, \tag{10}$$

where $D$ is the previously defined dimensionless phase diffusion constant in the absence of the external signal. For time $t \ll t_e$, the

phase variance follows diffusion $\langle \delta\psi^2 \rangle = Dt/\tau$. For $t \gg t_e$, the phase variance reaches a constant $\sigma_\psi^2 = Dt_e/\tau$. From Eq. (10), it is clear that phase fluctuation is suppressed by reducing $D$ even in the presence of external driving signal.

By using the Gillespie algorithm[39], we also computed the normalized auto-correlation function: $C_{xx}(t) = \langle x(t+s)x(s) \rangle_s / \langle x^2 \rangle = \exp(-\sigma(t))\cos(\Omega t)$, where $x$ is a state variable. As shown in Fig. 4c, the amplitude of a typical autocorrelation function $C_{xx}(t)$ first decreases exponentially before reaching a steady state constant $A = \exp(-Dt_e/\tau)$, which verifies Eq. (10). Based on the value of $A$ from the simulations, we obtain the phase variance $\sigma_\psi^2$, which gives $D = \sigma_\psi^2\tau/t_e$. As shown in Fig. 4d, the dimensionless diffusion constant $D$ decreases with energy dissipation $\Delta W$ consistent with the case without external signal (Fig. 3b).

**Energy-enhanced phase-amplitude coupling leads to higher phase sensitivity**. To understand the dependence of phase sensitivity on free-energy consumption analytically, we study the normal form of limit cycle oscillation originated from a Hopf bifurcation. Applying variable transformation, we obtain from CLE the stochastic Stuart–Landau equation[40]

$$\frac{dr}{dt} = \mu r - \beta_1 r^3 + \frac{q_r}{\sqrt{V}}\xi_r(t)$$
$$\frac{d\theta}{dt} = \omega + \beta_2 r^2 + \frac{q_\theta/r}{\sqrt{V}}\xi_\theta(t) \tag{11}$$

where $\mu, \omega, \beta_1, \beta_2$ are parameters associated with original reaction rate constants and $\xi_{r,\theta}$ are unit variance white noise terms. We note that $\beta_2$ characterizes the phase-amplitude coupling strength that affects the phase dynamics. By using the stochastic averaging

method[41], we can show the two noise strength $q_r$ and $q_\theta$ to be the same $q_r^2 = q_\theta^2 \equiv Q$ to the leading order in amplitude. See Supplementary Note 6 for detailed derivations.

For $\mu > 0$, we have the mean amplitude $r_s = \sqrt{\mu/\beta_1}$ and angular velocity $\Omega = \omega + \beta_2 r_s^2$. The isochron with the phase $\theta$ can be determined from the mean-field solution (see Supplementary Note 7 for details)

$$\phi(r, \theta) = \theta - \frac{\beta_2}{\beta_1}\left(\ln r - \frac{1}{2}\ln\frac{\mu}{\beta_1}\right) \quad (12)$$

Thus, $\nabla_x \phi = \left(-\frac{\beta_2}{\beta_1}/r\right)e_r + (1/r)e_\theta$ and the phase sensitivity can be calculated

$$\chi = \sqrt{(\beta_2/\beta_1)^2 + 1}. \quad (13)$$

The dissipation of Stuart-Landau oscillators can be determined by solving the Fokker–Plank equation for (11)[19], which results in the steady state probability distribution

$$P^{ss}(r, \theta) = A \exp\left[-\frac{(\beta_1 r^4/4 - \mu r^2/2)}{\Delta}\right] \quad (14)$$

where $\Delta = Q/V$ and $A$ is the normalization coefficient. To have a well-defined phase variable requires that the amplitude fluctuation is much smaller than $r_s^2$, leading to a constraint $\rho \equiv \mu/\sqrt{2\beta_1\Delta} \gg 1$[19].

The entropy production rate computed from $P^{ss}(r, \theta)$ yields the minimum free-energy cost per cycle[42] ($k_BT = 1$, see also Supplementary Note 8 for details).

$$\Delta W = \dot{S} \times \frac{2\pi}{\langle\Omega\rangle} = \frac{2\pi}{\Delta} \times \frac{\langle r^2(\omega + \beta_2 r^2)^2\rangle}{\langle(\omega + \beta_2 r^2)\rangle}$$
$$\approx 4\pi\rho^2(1 + \kappa)\left(\frac{\beta_2}{\beta_1}\right) \quad (15)$$

where $\kappa = \omega/\beta_2 r_s^2$ is a dimensionless parameter.

Finally, assuming that the phase sensitivity is dominated by the radial contribution, i.e., $\beta_2/\beta_1 \gg 1$, we have $\chi \approx \beta_2/\beta_1$ and $\kappa \ll 1$. Inserting this into Eq. (15), we obtain that to the leading order:

$$\chi \approx K_W \Delta W, \quad (16)$$

where $K_W \approx [4\pi\rho^2]^{-1}$ is a constant independent of $\beta_2$.

The linear relation (16) agrees with our numerical results. More importantly, the analytical results reveal that the amplitude-phase coupling constant $\beta_2$ has an important role in relating energy dissipation (Eq. (15)) to phase sensitivity (Eq. (13)). We further develop a toy model of limit cycle oscillator (see Supplementary Fig. 6) where $\beta_2$ can be explicitly calculated in terms of the microscopic nonequilibrium parameter $\gamma$. See Supplementary Note 9 for details. We found, in this simple model, that free-energy cost of increasing forward rates and suppressing backward rates both strengthen the phase-amplitude coupling $\beta_2$, therefore enhancing the phase sensitivity.

**Design principles for enhancing oscillation functions.** Now that we show it is in principle possible to increase phase sensitivity and to suppress phase diffusion simultaneously in a nonequilibrium system that consumes free energy, the next logical question is what are the design principles for a biochemical oscillator to optimize desirable oscillatory behaviors with a fixed energy budget. Here we search for possible design principles by studying the performance of an oscillator,

characterized by $\chi$ and $D$, for different combinations of $\gamma_1$ and $\gamma_2$ that lead to the same free-energy dissipation per period. In particular, we look for rules for designing the kinetic rate parameters in reactions that form a microscopic loop like the one shown in Fig. 2c. Unlike in simplified models[43] where only one reversible reaction exists between two states, these microscopic loops, which are formed by two distinct reaction pathways between two nearby microscopic states, are the basic building blocks in realistic oscillatory biochemical networks.

**Balance the forward-to-backward ratio in antiparallel pathways to suppress phase diffusion.** Consider $N$ discrete states with equally spaced phases $\phi(n) = 2\pi n/N$ along a limit cycle, we study the case in which there are two antiparallel pathways $i = 1, 2$ between two neighboring states. For each pathway, its forward and backward transition rates are given by $w_i^{\pm}$, respectively. The overall effect of the $i^{th}$ transition is denoted by the net rate $w_i(\phi) = w_i^+ - w_i^-$. These two pathways are antiparallel in the sense that $w_1^+$ and $w_2^+$ are in the opposite direction and they together form a counter-clockwise loop (so do $w_1^-$ and $w_2^-$ in the clockwise direction; see Fig. 2c for the Brusselator model).

It is easy to show that the average change of phase and its variance are[44]:

$$\frac{d\langle\phi(t)\rangle}{dt} = w_f(\phi) - w_b(\phi) \equiv \Omega_N, \quad (17)$$

$$\frac{d\langle\delta\phi^2(t)\rangle}{dt} = \left(\frac{2\pi}{N}\right)^2 \sum_{i=f, b; \nu = \pm} w_i^\nu, \quad (18)$$

where the subscript $f, b$ can be either $i = 1$ or $i = 2$ depending on whether a given transition pathway $i$ contributes to forward or backward movement of the phase. As shown in Fig. 2c for the Brusselator model, a given reaction pathway, either the $B + X \rightleftharpoons D + Y$ reaction ($i = 1$) or the $2X + Y \rightleftharpoons 3X$ reaction ($i = 2$), can lead to a forward or a backward phase change depending on which part of the limit cycle (or phase) the transition occurs. For example, $f = 1, b = 2$ at phase $\phi = 0$, whereas $f = 2, b = 1$ at phase $\phi = \pi$ (see Fig. 2c).

From the above Eqs. (17) and (18), we obtain a dimensionless phase diffusion constant

$$D = \left(\frac{2\pi}{N}\right)^2 \sum_\phi \sum_{i=f, b; \nu = \pm} \frac{w_i^\nu}{\Omega_N}\bigg|_\phi. \quad (19)$$

Our goal is to minimize phase diffusion under the constraint of a fixed energy dissipation $\ln(w_1^+ w_2^+/w_1^- w_2^-) = \ln\gamma^{-1} \sim \Delta W/N$. For simplicity, we assume the ratio $w_i^+/w_i^-$ take the same value independent of the phase, i.e., $w_i^+(\phi)/w_i^-(\phi) \equiv g_i$. The fixed energy constraint is now given by $g_1 g_2 = \gamma^{-1}$. By using $w_f(\phi) = w_b(\phi) + \Omega_N$ and the expressions for $w_i^\pm$: $w_i^+ = w_i g_i/(g_i - 1)$, $w_i^- = w_i/(g_i - 1)$, we can rewrite diffusion constant as

$$D = \left(\frac{2\pi}{N}\right)^2 \sum_\phi \frac{g_f + 1}{g_f - 1}\bigg|_\phi + \left(\frac{g_f + 1}{g_f - 1} + \frac{g_b + 1}{g_b - 1}\right)\frac{w_b(\phi)}{\Omega_N}. \quad (20)$$

For an ideal symmetric clock, we can assume $g_b|_\phi = g_f|_{\phi+\pi}$, and the summation of the first term in Eq. (20) can be expressed as $\sum_\phi \frac{g_f+1}{g_f-1}\big|_\phi = \frac{N}{2} \times \left(\frac{g_1+1}{g_1-1} + \frac{g_2+1}{g_2-1}\right)$, which leads to a new expression

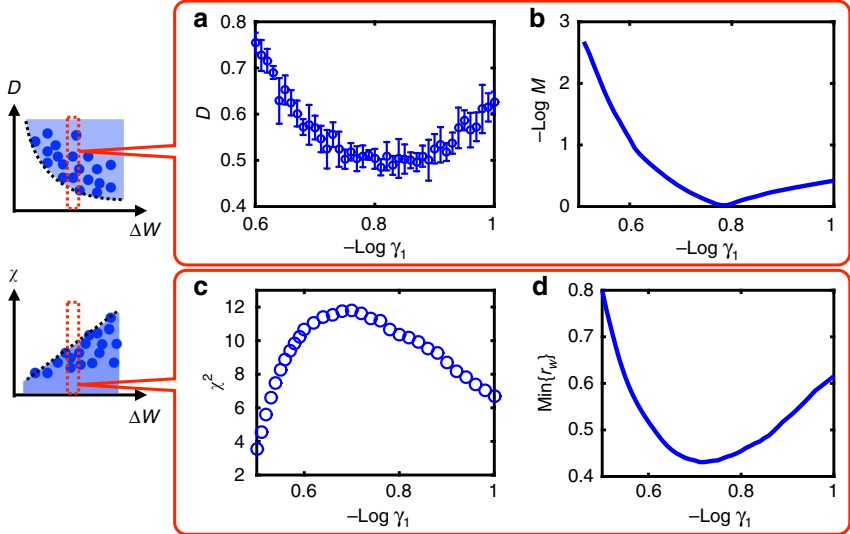

**Fig. 5** Testing the design principles in the Brusselator model. We vary $\gamma_1$ and $\gamma_2$ together with a fixed energy budget ($\Delta W = 200$). **a, b** The minimum phase diffusion $D$ occurs at $-\log \gamma_1 \approx 0.8$, where the forward-backward ratio is matched for the two antiparallel pathways, i.e., $M \approx 1$. **c, d** The phase sensitivity reaches its maximum at $-\log \gamma_1 \approx 0.7$, where $r_w$, the ratio of the net flux for the phase-retreating pathway over that of the phase-advancing pathway is minimum

for $D$ and consequently its lower bound:

$$D = \frac{(2\pi)^2}{2N}\left(1 + \frac{2}{N}\sum_\phi \frac{w_b(\phi)}{\Omega_N}\right) \times \left(\frac{g_1 + 1}{g_1 - 1} + \frac{g_2 + 1}{g_2 - 1}\right) \quad (21)$$

$$\geq \frac{(2\pi)^2}{N}\left(1 + \frac{2}{N}\sum_\phi \frac{w_b(\phi)}{\Omega_N}\right) \times \frac{1 + \sqrt{\gamma}}{1 - \sqrt{\gamma}} \quad (22)$$

where the lower bound is reached when $g_1 = g_2 = \gamma^{-1/2}$. Here we have implicitly assumed that the average of normalized backward rate $N^{-1}\sum_\phi w_b(\phi)/\Omega_N$ has weak dependence on $g_1$, $g_2$.

Our analysis clearly suggests that one design principle to minimize the phase diffusion of biochemical oscillators with a given energy budget is to equalize (balance) the ratio between forward and backward transition rates in the two antiparallel pathways that form the (non-equilibrium) dissipative loop.

This design principle obtained from an ideal clock is consistent with our previous findings in the AI model[19]. We have also directly tested it by numerical simulations of the Brusselator model, where two adjacent states $(X - 1, Y + 1)$ and $(X, Y)$ are linked by a non-equilibrium reaction loop (see Fig. 2c). We compute the ratio $r$ of the forward and backward flux of two reactions, respectively: $r_1 = w_1^+/w_1^- = k_2[X]/k_{-2}[Y]$ and $r_2 = w_2^+/w_2^- = k_3[X]^2[Y]/k_{-3}[X]^3$. We can then define $M \equiv \max\left(\langle r_1(t)\rangle_\tau, \langle r_2(t)\rangle_\tau\right)/\min\left(\langle r_1(t)\rangle_\tau, \langle r_2(t)\rangle_\tau\right)$ to measure the matching of these two pathways averaged over the period. As shown in Fig. 5a, b, the exact matching point ($M = 1$) correlates closely to the minimum of the phase diffusion (see Supplementary Figs. 7 and 8 for other values of $\Delta W$).

**Minimize the net backward flux relative to the net forward flux to enhance phase sensitivity.** To look for strategies in enhancing phase sensitivity, we calculate $\| \nabla_{x^*}\phi \|$ by projecting the transition fluxes in the state space onto the phase (angular) direction and the amplitude (radial) directions. In addition to the reactions that form part of the loop (such as $w_1$ and $w_2$ shown in Fig. 2c), we also include the reaction that is not part of the dissipative loop, e.g., the $A \rightleftharpoons X$ reaction in the Brusselator model. The net

flux for this non-dissipative reaction is denoted by $w_0$, which is not regulated by the non-equilibrium parameters ($\gamma_{1,2}$).

Recall that phase sensitivity $\chi$ measures the largest $\| \nabla_{x^*}\phi \|$ along the limit cycle. It can be shown that the maximum phase gradient occurs at the point in phase space where the energy-driven fluxes most closely align with the phase direction (see Supplementary Note 10 for a more detailed analysis). A corresponding approximation is that the transition rate in the radial direction near the most sensitive point is controlled by $w_0$. Thus, we can estimate $\chi$ as:

$$\chi \approx \frac{w_f}{w_0} \times \frac{|q_f - (w_b/w_f)q_b|}{q_0} \equiv c_w\frac{|q_f - r_wq_b|}{q_0} \quad (23)$$

where $q_i = \frac{\Delta w_i/w_i}{\Delta r/r_s}$ is the relative change of transition rates induced by a perturbation given by a relative change $\Delta r/r_s$ of the amplitude. $q_i$ characterizes the sensitivity of the $i^{\text{th}}$ transition determined by the nonlinearity in the underlying reaction rates. In biochemical oscillators, for example, reactions of (pseudo) first order have $q_i \sim 1$; the autocatalytic reactions in the Brusselator model give $q_i \sim 3$.

The dominant effect of energy dissipation is to increase $w_f$, which roughly takes the form $c_1/\gamma_1 - \gamma_2 c_2$, where $c_{1,2}$ are concentration-related coefficients for biochemical circuits. At a fixed energy dissipation or a fixed $w_f/w_0$, Eq. (23) suggests that reducing the net backward flux relative to the net forward flux, i.e., minimizing $r_w \equiv w_b/w_f$ can lead to higher phase sensitivity.

This design principle, based largely on heuristic arguments, has been tested directly in the Brusselator model where the two net fluxes are $w_1 = k_2[X] - k_{-2}[Y]$ and $w_2 = k_3[X]^2[Y] - k_{-3}[X]^3$. We calculated $r_w = \min(w_1, w_2)/\max(w_1, w_2)$ numerically along the entire limit cycle and compared $\min\{r_w\}$ for different combinations of $\gamma_1$ and $\gamma_2$ for a fixed dissipation $\Delta W$. As shown in Fig. 5c, d, phase sensitivity reaches its peak when $r_w$ is small, which confirms the strategy (design principle) for maximizing phase sensitivity by a maximum separation of net transition fluxes between anti-parallel pathways in a dissipative loop. Similar results were confirmed in the AI model (see Supplementary Fig. 9 for details).

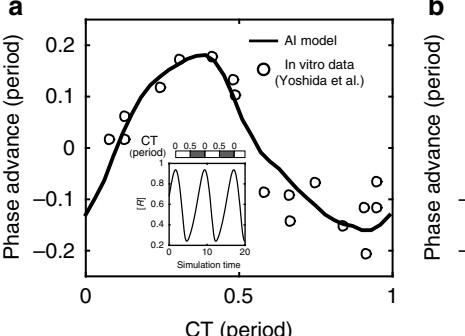
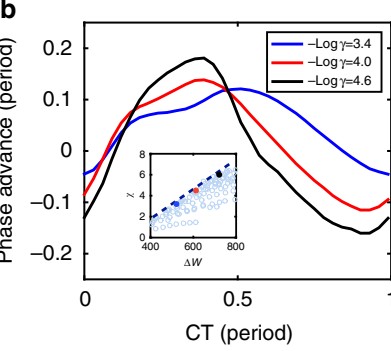

**Fig. 6** Experimental connection and prediction of the dependence of phase sensitivity on irreversibility $\gamma$. **a** By assuming $k_1$ is increased by 70% and $k_4$ by 30% during a temperature up perturbation (0.1 period), the PRC from the AI model (black line) can be fitted to that of a temperature up PRC (symbols) observed in[57]. Simulation trajectory of oscillation along with the circadian time scale are shown in the inset. The peak of activator rhythm is defined as CT 0. Experimental PRC data from ref. [57] is aligned with the model PRC to peak at the same CT. **b** PRCs for the same perturbation as in **a** is computed for different background levels of ATP/ADP ratio or equivalently different values of $\gamma$ (given in the legend). The extra free-energy dissipation for each $\gamma$ choice are: $\Delta W - W_c \approx 170$ (blue); 250 (red); 350 (black, same as in **a**). Amplitude of PRC increases as sensitivity $\chi$ is enhanced by $-\log\gamma$ or $\Delta W$ as shown in the inset, where the hollow circles correspond to different choices of $(\gamma_1, \gamma_2)$. This prediction can be tested experimentally

**Experimental evidence in the Kai system**. To look for experimental evidence of the design principles described above, we investigate the circadian clock of cyanobacteria, i.e., the Kai system, by studying the experimentally measured kinetic rates in the reactions between different phosphorylation states ($U$, $T$, $D$, $S$) of KaiC along its PdP cycle[45–47].

To test the first design principle, we calculate the ratio of forward to backward flux $r_{X \rightleftarrows Y} = J_{X \rightarrow Y} / J_{Y \rightarrow X}$ on each of the four links in the KaiC phosphorylation cycle $U \rightarrow T \rightarrow D \rightarrow S \rightarrow U$ based on a kinetic model of KaiC phosphorylation with experimentally measured rate parameters[45–47] (see Supplementary Note 11 and Supplementary Fig. 10a for details of the model and Supplementary Table S1 for the parameters). We then compare the period-averaged ratio $\langle r_{X \rightleftarrows Y} \rangle_\tau$ to check whether these ratios are matched along the cycle. The KaiC model with the experimentally determined parameters (see Supplementary Information for details) yields $\langle r_{U \rightleftarrows T} \rangle_\tau : \langle r_{T \rightleftarrows D} \rangle_\tau : \langle r_{D \rightleftarrows S} \rangle_\tau : \langle r_{S \rightleftarrows U} \rangle_\tau = 1 : 1.16 : 0.97 : 1.08$, clearly showing that the forward-to-backward ratio in different links (pathways) of the KaiC phosphorylation cycle are properly balanced, which is consistent with the first design principle for minimizing phase diffusion.

To test the second design principle, we calculate the net flux for each link $J_{X \rightarrow Y}^{(net)} = J_{X \rightarrow Y} - J_{Y \rightarrow X}$. The net phosphorylation and dephosphorylation flux are then approximated as $J_{U \rightarrow D}^{(net)} = J_{U \rightarrow T}^{(net)} + J_{T \rightarrow D}^{(net)}$ and $J_{D \rightarrow U}^{(net)} = J_{D \rightarrow S}^{(net)} + J_{S \rightarrow U}^{(net)}$, respectively. From direct simulation of the model, the backward-to-forward (in terms of phosphorylation rhythm) net flux ratio $r_w$ is smaller during the subjective day with an average $\langle r_w \rangle_{day} \approx 0.2$ than during the night with average $\langle r_w \rangle_{night} \approx 0.5$. According to the second design principle, this $r_w$ behavior leads to a higher phase sensitivity during the subjective day (phosphorylation phase) than during the night (dephosphorylation phase), which is consistent with the experimentally measured PRC reported in ref. [46] (see Supplementary Note 11 and Supplementary Fig. 10 for details). This result confirms that the Kai system controls phase sensitivity by modulating the relative strength of the phosphorylation and dephosphorylation fluxes.

## Discussion

In this study, we investigated whether and how biochemical systems can achieve high sensitivity and low fluctuation at the same time in the context of biological oscillators. In non-equilibrium systems, the FDT is broken. There is no unique relationship between fluctuation and response—they could have a

positive correlation or a negative one depending on which parameters are varied. For example, in the Stuart–Landau model, if we vary a single variable without changing the others, $\chi$ and $D$ satisfy a positively correlated relation: $\chi^2 = D/T_{eff}$. However, even in this case, the effective temperature $T_{eff}$ could be lowered by dissipation $\Delta W$ in different ways depending on which variable ($\mu$ or $\beta_2$) is varied (see Supplementary Note 12 for details). For a realistic non-equilibrium biochemical reaction network (circuit) such as the Brusselator model, where many parameters are affected by the dissipation in a correlated manner, our study shows that an increase in free-energy dissipation can lead to both a suppression of phase fluctuation and an enhancement of the phase sensitivity if it is used properly (or the circuit is designed properly). This result should be generally applicable to other non-equilibrium systems.

Our study revealed two design principles to achieve optimal performance with a finite budget of free-energy consumption in oscillatory systems. The findings of our analysis can be demonstrated by the expression of dissipation rate $\dot{W} = \sum_i w_i \ln(w_i^+ / w_i^-)$. In principle, high sensitivity and low phase fluctuations for a given dissipation rate can be achieved by favoring the net forward flux $w_f$ over the net backward flux $w_b$ and balancing the forward-to-backward ratio $w_i^+ / w_i^-$ among antiparallel reaction pathways, respectively. As these two design principles act on different combinations of the reaction rates, they can be satisfied simultaneously leading to biochemical circuits that have both high sensitivity and low fluctuations. Strong evidence in support of these two design principles are found in the Kai system. Aside from helping us understand the structure and dynamics of naturally occurring biological pathways, these design principles may also serve as the best practice rules for constructing efficient synthetic biochemical circuits for oscillations.

Other evidence for our theory may be found in experiments measuring PRC at different energetic states of the system. For example, our theory predicts that the PRC amplitude should decrease when the background ATP/ADP ratio is decreased. As far as we know, no such experiment has been done. The closest are PRC measurements (to light and drug/chemical perturbations) at different temperatures. In our analysis, the dimensionless energy dissipation $\Delta W$ has a temperature-dependent component $\beta \Delta G^{(0)}$, where $\beta = (k_B T)^{-1}$ is the inverse thermal energy and $\Delta G^{(0)}$ is a concentration-independent free-energy difference[48]. According to Eq. (6), this means the phase sensitivity increases as the temperature decreases if we neglect other

temperature dependence. Interestingly, in the experimental systems we found, i.e., Neurospora[49], Gonyaulax polyedra[50], and chick pineal cell[51], except for the case of light response in chick pineal cell where no $T$-dependence is found, PRC amplitudes indeed increase with $1/T$ (see Supplementary Fig. 11). However, many internal variables in biological systems can depend on temperature, therefore these temperature dependence measurements in in-vivo systems may not serve as direct tests of our theory.

The best system to test our theory directly is the relatively simple cyanobacterial circadian clock, especially the in vitro Kai system. Experimental studies in cyanobacterial circadian clocks uncover that metabolism is the fundamental synchronizer[46,47,52]. However, there is yet no direct experiments investigating the relationship between phase sensitivity and the background metabolic state of the system. Away from the onset of oscillation, the relative amplitude change $\Delta r/r$ caused by a modest phase resetting signal (e.g., temperature pulse) should has only a weak dependence on the metabolic state of system. Under this assumption, our theory shows that the phase response is dominated by $\chi$, which increases with the background ATP/ADP ratio. In particular, we predict that the invitro KaiABC oscillators with a lower background ATP/ADP ratio would generate a smaller PRC amplitude in response to the same temperature perturbation. In Fig. 6, we show the PRC in response to a particular change of kinetic rate constants in a generic AI model, to which the Kai system belongs. By choosing the reaction rate changes during the temperature perturbation properly, we can reproduce the observed PRC to a temperature pulse as shown in Fig. 6a. We then use our model to compute the PRC to the same perturbation (rate changes) but with different values of the energy dissipation per period ($\Delta W$) or different background ATP/ADP ratios. As shown in Fig. 6b, the amplitude of the PRC decreases as $\Delta W$ decreases. This result is robust as the enhancement of PRC amplitude by free-energy dissipation holds true generally regardless of the specific perturbation and the specific model used (see Supplementary Fig. 12). This prediction can be tested in future experiments, e.g., by measuring the PRC for changes in temperature in different background ATP/ADP ratios. These types of experiments would not only test our hypothesis that energy dissipation affects phase response, they may also shed light on its molecular mechanisms[53] and how cellular metabolic activities affects other crucial functions of biological clocks, such as temperature compensation[54–56].

**Data availability**. The data that support the findings of this study are available from the corresponding author on request.

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

## Acknowledgements

The work is supported by Chinese Ministry of Science and Technology (2015CB910300) and NSFC (11434001). Y.T. is also supported by NIH (GM081747).

## Author contributions

Y.T. initiated the project. C.F., Y.C., Q.O., and Y.T. designed the research. C.F., Y.C., and Y.T. developed the models, contributed to the analytical results, and wrote the manuscript. C.F. performed simulations.

## Additional information

**Competing interests:** The authors declare no competing interests.

