## [Peer Review File · Nature Communications]

Reviewers' comments:

Reviewer #1 (Remarks to the Author):

In this manuscript Fei et al. analyze a handful of models with biochemical oscillations and find that biochemical oscillators can get more precise and increase their sensitivity with an increase in energy dissipation. I have an extended list of points for the authors to address. As a summary of my critique: I am not sure whether the results presented in this paper are universal as there are simple examples of systems that get "less precise" with an increase in energy dissipation; their definitions only work for biochemical oscillators that display a limit cycle in the deterministic limit and there are examples of biochemical oscillators that do not display such limit cycle; at least one of their "design principles" has already been found in the literature.

1) The definition of sensitivity and of phase diffusion seem to depend on the existence of limit cycle within the deterministic rate equation description. However, biochemical oscillations in stochastic systems can happen also in models that do not display a limit cycle in the deterministic limit (see A. J. McKane et al, J. Stat. Phys. 128, 165 (2007))? How do you define these quantities in such cases? For a general master equation (or chemical master equation), what is the definition of phase diffusion? What is the relation between phase diffusion and the coherence time τ_c for a general master equation?

2) Just as an example, in Ref. 19 the authors consider a single cycle model towards the end of this reference. For such single cycle model (Fig 6 in the arXiv version of the paper) with arbitrary site dependent rates, what is the definition of phase diffusion, sensitivity and coherence time for this simple model? What is the relation between phase diffusion and coherence time in this model? The authors may restrict to the case $N=3$ states to simplify the discussion.

3) It seems that the first “design principle” found by the authors is not new. In A. C. Barato and U. Seifert, Phys. Rev. E 95, 062409 (2017), there is a similar result (appendix A) showing that the precision of biochemical oscillators for a given “energy budget” is maximized by setting the kinetic parameters with an even distribution of the parameter γ among the different transitions.

4) I think the results in this paper depend on kinetic parameters. There are models where biochemical oscillations can get less precise with the increase in energy dissipation. For such models there is an optimal amount of energy dissipation. In A. C. Barato and U. Seifert, Phys. Rev. E 95, 062409 (2017), there are examples of models that contradict the behaviour found in Ref. 19. How universal are the results of this paper? What are the restriction on the kinetic parameters for the increase in energy dissipation to imply an increase in precision and sensitivity in biochemical

oscillations? It does not seem to be hard to imagine a simple three state model where the “sensitivity” would also show an optimal value as a function of the energy dissipation.

5) Could the authors fully define the models in the supplement. For example, for the AI model they simply wrote down the deterministic rate equation. What kind of numerical method the authors used for this model? What is the chemical master equation for this model? The model seems to have irreversible reactions like $X+R \rightarrow X$, which would lead to infinite energy dissipation according to formula (4). Could you clarify this point?

6) There are examples of really bad writing in the supplement. The first paragraph of section SVI that contains equation S15 is one such example. The authors have to improve the supplement a lot. Some parts are really hard to follow.

Overall, there is nothing particularly new on the fact that it is possible to find models where both precision and sensitivity improve with an increase in energy dissipation. The main question is: how universal this behaviour is? In other words, under which conditions on the kinetic parameters?

Reviewer #2 (Remarks to the Author):

In this manuscript, the authors proposed designing mechanisms for certain types of biological oscillators to increase both its sensitivity and robustness by varying the driven forces of free energy dissipations. This design is not allowed in the linear response regime but becomes possible in systems driven far away from equilibrium.

The work is technically interesting. The question formulated in the abstract and introduction is an important one. Their result is in terms of forward and backward fluxes which seem like nice variables to use.

However, there are two serious problems with the work that raise questions both about its validity and its relevance and thus prevent recommending this paper for publication:

Biological oscillatory systems studied here are externally driven oscillators. But the authors define phase sensitivity and phase diffusion, the two central quantities of this paper and several others, in the absence of a driving signal. Many publications have shown that the results are very different when externally driven.

The experimental connection shown in Fig 5 is a pretty far out interpretation of the experimental data. The paper would almost be stronger without it.

Without experimental data, the paper could have been an interesting theoretical exercise in computing non-equilibrium steady state properties, suitable for specialized journals. But (1) above is a serious design flaw in even metaphorically linking the analysis to the biological systems the authors mention (or indeed any imagined context). Unless the authors can derive results for oscillators with external driving - the putative subject of the paper - one cannot recommend this manuscript for publication in an interdisciplinary journal.

The above criticisms are detailed below:

1. Externally forced vs unforced systems:

As is widely appreciated, biological systems need to respond to relevant 'external cues' in environmental signals while also ignoring all kinds of irrelevant fluctuations within. The common sense way of modelling this is to define a quantity that quantifies each of these.

Here, the authors choose to study oscillatory systems like circadian clocks and the authors define a phase sensitivity χ for the former and a phase diffusion constant D for the latter concept.

However, the definition of both quantities in this paper is quite problematic. These oscillators are always used in the context of an externally time varying signal that entrains the system. The 'external cues' could then be e.g., changes in amplitude or phase of that signal. A commonly studied example would be how these oscillators overcome 'jet lag' (i.e., how they quickly respond to phase changes of the external signal).

Instead, in the authors' calculations, there apparently is no driving signal at all. It is as if the 'external cues' is the presence of an external signal in the first place.

This issue is important both for technical reasons and for motivational reasons. Technically, over the last 20 years, most papers on this subject have noted phase diffusion is very different in externally driven systems vs undriven systems - see papers by Gonze, Goldbeter etc. For more recent work on this topic, see recent work in Cell Systems from the Khammash group. Similarly, ten Wolde's group has a recent work showing that driven systems have very different phase diffusion from the undriven system modelled here.

In terms of motivation for the paper, it is also hard to imagine any situation for oscillators that operates without any entraining signal present all the time. In fact, the authors clearly recognize this since they define entrainability as a critical aspect (e.g., in the SI) and talk about synchronizability and about Arnold tongues in different parts of the paper. And yet all the calculations of phase diffusion were done without any periodic external drive that's clearly present in all the examples mentioned in the paper and in the experimental systems this manuscript connects to.

Perhaps the authors have restricted themselves to the case of no external signal because, with an external signal, the question of dissipation becomes quite academic. E.g., the external signal may also do work on the system and it is not clear this current kind of energy budgeting is relevant.

In summary, it is hard to say what real question this calculation is directly relevant for, especially at the level required for journals like this. Further, existing works show why ignoring the ever-present periodic external signal is a bad idea in defining phase diffusion.

2. Experimental data:

The authors rely on two kinds of possible experimental support to their theory - see Fig 5. However, neither panel shows a check of the theory.

Fig 5a: The authors first look at the temperature dependence of the phase response curve (PRC) in *Neurospora* and a couple of other organisms. The data shows that the size of the PRC generally increases with temperature but the authors want to conclude a lot more - that these relationships are linear and that their slopes are the same.

To start with, two of the organisms have only two data points! Further, the x-axis has an extremely small range (for good reason as we'll discuss later) and there are huge error bars in y, makes the slopes of these curves meaningless. To conclude that these relationships are linear and that their slopes are the same really requires a very very favorable reading of these results. Finally, a linear scaling is not enough to hang a hat on - even without any of the author's theory, many other simple assumptions would predict the same. E.g., see works on temperature compensation in clocks.

A more serious flaw is that the experiments in panel (a) were performed on living organisms. The effect of temperature on the phase response of a living organism can in no way be traced to the core oscillator's properties alone! There is a reason the x-axis of Fig 5a has such a tiny range - the organisms would die for the kind of serious temperature changes that would actually test the author's physical theory. To put another way, there is way too much biology going on in the temperature response data shown in Fig 5a to compare it to a Boltzmann factor.

For example, Ref [45], from whom the PRC data on chick pineal cells was obtained, explicitly states that the change of the PRC results from an increase in the size of the limit cycle with temperature. Such larger amplitude oscillations result from multiple biological factors as discussed in the Discussions of that paper - it is hard to imagine how one could ascribe all or even most of the effect of such a temperature change to the kind of simple Boltzmann factor analysis done here.

Similarly, Ref [46], from which PRC data on dinoflagellates were obtained, states that the PRCs due to light actually showed no temperature dependence! The authors here use the drug PRC data from the paper which does show temperature dependence, showing conclusively that the PRC's temperature dependence is not a property of the oscillator itself (the subject of the author's study) but because of the many upstream and downstream temperature-sensitive processes. Indeed, Ref

[46] discusses many such explanations and tries to quantify their contributions - e.g., drugs affect protein synthesis differently at different temperatures while light does not, temperature-sensitive phase angle between overt rhythm and the pacemaker phase etc. It seems hard to conceive of a situation where all of these strong obvious temperature effects are somehow less important than the subtle temperature dependence of the oscillator itself.

I completely understand and support the author's desire to build a solid physically grounded theory of one piece of the cellular system. We certainly need more of such theory, and the above criticism should not be a mistaken for a criticism of making modelling assumptions or studying subparts of a complex system. The problem is in comparing to two data points of whole organismal experiments to such a theory, when there are clearly much stronger biological effects at play.

Finally, the putative connection to the Kai system suffers a different serious weakness. Several papers, e.g., from O'Shea lab which the authors cite, have shown experimentally that the size of the PRC is primarily set by the difference in ATP levels between day and night. The authors' theory here links the size of the PRC to the absolute ATP levels, say, in the day. But all these other publications - from the time of Arthur Winfree's book - have shown the strong impact of the difference between the day and night cycles on the height of the PRC.

Again, it is hard to imagine a situation where the change in PRCs can be attributed to the absolute ATP levels when the effect of difference in ATP levels has already been documented to be the main dominant effect. In fact, Winfree's famous classification of type 0 and type 1 PRCs is an example of how strong the latter effect is. (For general theory on this, one could consult Winfree's book or Pikowsky's book titled 'Synchronization'. For a recent experiment on the Kai system satisfying this property, see the recent Molecular Cell paper from the O'Shea lab.)

Reviewer #3 (Remarks to the Author):

Fei et al. present a detailed analysis of how energy dissipation in oscillating reaction network is related to phase sensitivity and phase diffusion behavior. The work follows up on a previous work by authors which studied phase diffusion in oscillatory biochemical reactions. The current work mainly focuses on the relation between energy dissipation and phase sensitivity.

The relation is illustrated on several simple models, but a general relation between the amount of dissipated energy and phase sensitivity is derived for all oscillatory circuits.

I consider the work to be innovative and important, the results are sound and make testable predictions. In the present form, I would find the work suitable for PRX

or similar journal, but for the broad audience of Nature Communications, I would ask for expansion of the part of the article that makes explicit connections with known biological oscillators.

In particular, it is not clear to me that phase sensitivity is a general important feature for biological circuit. For instance, the circadian oscillations are

synchronized by a long term exposure to daylight, rather than by a short perturbation. While the author's results on the relation between dissipated energy and phase sensitivity holds in general, it might

not be the case that circuits are designed for achieving best phase sensitivity for a given amount of dissipated energy. In section C, predictions are made for the circuit properties that would optimize

phase sensitivity / phase diffusion for a given amount of energy W . Can a comparison be made to parameters of known biochemical oscillator, such as the examples used in Fig 5a?

Further points:

1) The AI model is one of the main models used to illustrate the theoretical results, and should be hence presented in main text too (Figure S1). Various combinations of values of parameters γ_1 and γ_2

are studied, I would be curious to include in the plot the typical values of $\gamma = \gamma_1 / \gamma_2$ (ie plugging into the gamma formula the typical concentrations of ATP, ADP, P and K_{eq}) encountered in PdP cycle

in living systems.

2) In the Brusselator model, why is k_1 and k_{-1} not included in the forward/ backward rates?

3) The biological oscillators studied in Fig 5 should be briefly introduced and described in Supplementary Information

4) In the title of the article, I would suggest replacing "biochemical networks" by "biochemical oscillatory systems", as this more accurately describes the studied system

5) Fig 5b and S9 captions should also include description of blue data points in the inset

Point-by-Point Responses to Comments from Reviewer #1

Reviewer #1 (Remarks to the Author):

In this manuscript Fei et al. analyze a handful of models with biochemical oscillations and find that biochemical oscillators can get more precise and increase their sensitivity with an increase in energy dissipation. I have an extended list of points for the authors to address. As a summary of my critique: I am not sure whether the results presented in this paper are universal as there are simple examples of systems that get “less precise” with an increase in energy dissipation; their definitions only work for biochemical oscillators that display a limit cycle in the deterministic limit and there are examples of biochemical oscillators that do not display such limit cycle; at least one of their “design principles” has already been found in the literature.

1) The definition of sensitivity and of phase diffusion seem to depend on the existence of limit cycle within the deterministic rate equation description. However, biochemical oscillations in stochastic systems can happen also in models that do not display a limit cycle in the deterministic limit (see A. J. McKane et al, J. Stat. Phys. 128, 165 (2007))? How do you define these quantities in such cases? For a general master equation (or chemical master equation), what is the definition of phase diffusion? What is the relation between phase diffusion and the coherence time τ_c for a general master equation?

Response: The type of oscillations reviewer 1 referred to are the noisy oscillations caused by stochastic resonance, where there is no limit cycle in the deterministic dynamics and the oscillation is driven primarily by noise. However, our current study focuses on a large class of biochemical oscillations where limit cycles do exist in the underlying deterministic dynamics. These systems are critical for various biological functions that require precise time-keeping. Specific examples include various biological clocks (e.g., the Kai system in Cyanobacteria), cell cycle oscillations, and biochemical oscillations (e.g., the cAMP oscillation in Dicty. cells). In these systems, the noise is caused by the stochastic nature of the intrinsic biochemical reactions, which degrades the quality of the oscillations. The goal of our current study is to find out how nonequilibrium effects with energy dissipation can help increase (or maintain) the performance of the

oscillation, such as period accuracy and phase sensitivity, despite the presence of these large noises.

While it is interesting to extend the concepts in our current study to other cases including the stochastic resonance models, it is beyond the scope of our current study. However, we have included some general ideas/thoughts on this subject in our response to the next question.

Revision: To clarify the goal of our current study, we have now revised our manuscript to explicitly state that our focus is on systems with limit cycles in the underlying deterministic dynamics in the introduction section of the manuscript (page 1, right column).

2) Just as an example, in Ref. 19 the authors consider a single cycle model towards the end of this reference. For such single cycle model (Fig 6 in the arXiv version of the paper) with arbitrary site dependent rates, what is the definition of phase diffusion, sensitivity and coherence time for this simple model? What is the relation between phase diffusion and coherence time in this model? The authors may restrict to the case $N=3$ states to simplify the discussion.

Response: In general, we can determine the coherence time τ_c by computing the auto-correlation function $C(t)$ of the state variable, and fit it to $C(t) = e^{-t/\tau_c} \times \cos\left(\frac{2\pi t}{T}\right)$, where T is the period of the oscillation. Once τ_c is determined, the phase diffusion constant is then given by: $D_\phi = \alpha \times (2\pi)^2 / \tau_c$, where α is a constant dependent on the waveform of the oscillation. This is explained in detail in ref. 19 (Cao et al, Nat. Phys. 2015) for cases where a limit cycle exists in the deterministic equation with a well defined period T . For cases where there is no limit cycle in the deterministic equation, the same procedure can be carried out to determine the coherence time and phase diffusion constant with the only difference being that the “period” T needs to be determined by the peak of the power spectrum or the imaginary part of the eigenvalues of the Jacobian matrix of the stable fixed point. A paper by Qian and Qian (H Qian and M Qian, PRL, 1999) gave a thorough analysis for the simple 3-states model mentioned. In addition, the phase sensitivity can also be defined even when there is no stable limit cycle as long as a certain perturbation induces a change in the state space. We can measure the phase shift between the perturbed and

unperturbed stochastic trajectory by computing the cross-correlation function. A proper definition of phase sensitivity would then be the phase shift normalized by the perturbation strength.

3) It seems that the first “design principle” found by the authors is not new. In A. C. Barato and U. Seifert, Phys. Rev. E 95, 062409 (2017), there is a similar result (appendix A) showing that the precision of biochemical oscillators for a given “energy budget” is maximized by setting the kinetic parameters with an even distribution of the parameter γ among the different transitions.

Response: The first “design principle” we found here is for reducing phase diffusion with a given energy budget, which is a generalization and extension of what we found in our previous work, see Fig. 4 in the 2015 paper by Cao et al (Nat. Phys. 2015), where we defined an efficiency of suppressing phase diffusion and concluded from our simulation studies of the AI model that “This result indicates that high efficiency is achieved when the kinetic rates in the two halves of the PdP cycle (phosphorylation and dephosphorylation) are matched” as stated at the end of the caption of Fig. 4 in the 2015 Cao et al paper. Furthermore, as stated in 2015 Cao et al paper (page 4 bottom left): “In the Supplementary Information we show in a simple model of chemical reaction cycles that with a constant energy dissipation (that is, a fixed γ), phase diffusion approaches its minimum when the forward and the backward rates along different steps of the cycle, for example, the PdP cycle, are matched.” This result seems to be similar to the one found in the simple discrete model in the recent work (A Barato, U Seifert, PRE, 2017) mentioned by the reviewer.

However, despite their similarity, there are significant differences between the design principle found here and that studied in the Barato and Seifert paper. The major difference is caused by the fact that in the realistic models we studied (e.g., the Brusselator model and the activator-inhibitor (AI) model) the two chemical states (X and Y) are connected by two distinctive reversible chemical reactions (pathways) with four reaction rates (w_1^\pm, w_2^\pm) whereas in the simple discrete model used by Barato and Seifert (PRE 2017) the two states are only connected by two phenomenological transition rates. These two pathways may not be combined as they correspond to different biochemical reactions as illustrated in Fig. 2 in our manuscript, which explicitly shows the two antiparallel pathways between the two nearby states “X” and “Y” in the Brusselator model (here,

antiparallel simply means that the “+” directions of the two pathways are opposite to each other). The same goes for the AI model where the bidirectional phosphorylation pathway and the bidirectional dephosphorylation pathway need to be considered separately: the inverse of phosphorylation reaction of a protein by ATP involves the recombination of ADP with the phosphate group on the protein back to ATP, it is not the dephosphorylation reaction of the phosphorylated protein to its unphosphorylated form and an inorganic phosphate.

As illustrated in Fig. 2b for the Brusselator model, the existence of these two distinctive pathways leads to a local transition flux cycle between the two states characterized by an overall irreversibility parameter: $\gamma = g_1 \times g_2 = \frac{w_1^- w_2^-}{w_1^+ w_2^+} < 1$, where $g_1 = \frac{w_1^-}{w_1^+}$ and $g_2 = \frac{w_2^-}{w_2^+}$ are the ratios of the forward and backward reaction rates for the two antiparallel reaction pathways. For a given γ or equivalently a given energy dissipation rate, we ask the question on how to choose g_1 and g_2 to minimize the phase diffusion and optimize the phase sensitivity. The first design principle we found is that the phase diffusion is minimized when these two ratios are the same $g_1 = g_2 = \sqrt{\gamma}$. Therefore, it is clear that this design principle is new as it can only be discovered when the two distinctive reaction pathways connecting nearby microscopic states are considered explicitly as we did in this work. The same goes for the second design principle that leads to a heightened phase sensitivity with a fixed energy budget.

Overall, the two design principles identified in this work are new and only possible when realistic microscopic reactions are considered explicitly as we did in this work.

Revision: We thank the reviewer for pointing out this new paper by Barato and Seifert (PRE 2017), which we now cite in our revised manuscript. We have also included a statement to clarify the difference between the design principles found here and that found in this previous work in the beginning of the subsection on design principles.

4) I think the results in this paper depend on kinetic parameters. There are models where biochemical oscillations can get less precise with the increase in energy dissipation. For such models there is an optimal amount of energy dissipation. In A. C. Barato and U. Seifert, Phys. Rev. E 95, 062409 (2017), there are examples of models that contradict the behaviour

found in Ref. 19. How universal are the results of this paper? What are the restriction on the kinetic parameters for the increase in energy dissipation to imply an increase in precision and sensitivity in biochemical oscillations? It does not seem to be hard to imagine a simple three state model where the “sensitivity“ would also show an optimal value as a function of the energy dissipation.

Response: We agree that the performance of a specific biochemical oscillator with a given energy budget will depend on its specific kinetic rate parameters. In fact, this is exactly the motivation for us to answer the two related questions in this work: (1) what is the optimal performance (minimum phase diffusion and maximum phase sensitivity) for a given energy budget if one has the freedom to choose all the kinetic rate parameters in realistic biochemical networks? (2) what are the design principles for choosing the kinetic rate parameters to reach the optimal performance?

We believe that the results, which we presented here in our efforts to answer these two questions listed above, are universal as they apply to a wide set of different realistic biochemical oscillators where we showed that free energy dissipation indeed sets the upper bounds on the system’s ability to reduce phase diffusion and enhance sensitivity. As we clearly stated in our manuscript, energy dissipation is a necessary condition for enhancing performance in these nonequilibrium systems, but not sufficient. There are infinite ways to waste energy without enhancing performance. As can be seen from Fig. 3b&d in the current manuscript (or Fig. 4a in the 2015 Cao et al Nat. Phys. paper), there are cases where systems dissipate more free energy but perform worse on either precision or sensitivity. Indeed, how to choose the key kinetic parameters within a realistic biochemical network to reach the optimal performance for a given energy dissipation is exactly one of the key questions (question (2) listed above) we addressed in this work by uncovering the two new design principles that lead to the optimal performance for a given energy budget.

We do not know the exact reason for the non-monotonic dependence of the accuracy versus energy dissipation mentioned by the reviewer in a recent work. It could be caused by the constrained parameter space imposed by the specific (non-generic) network topology in an idealized model like that used in Barato and Seifert (PRE 2017). For example, the first design principle, which we found to be crucial for improving precision with a fixed

energy budget, cannot be fully satisfied if one limits the reaction network to be that there is only one reversible reaction connecting two nearby states in such a simplified model (see our response to the previous question #3).

What we did show in this work is this. In ALL the realistic models (the AI models, the Brusselator model, different models for the kai system, etc...) we studied where all distinctive pathways between two states are considered, when we search the parameter space thoroughly, i.e., allowing all key rate parameters to vary independently, the optimal performance always increases and eventually saturates with the energy dissipation.

5) Could the authors fully define the models in the supplement. For example, for the AI model they simply wrote down the deterministic rate equation. What kind of numerical method the authors used for this model? What is the chemical master equation for this model? The model seems to have irreversible reactions like $X+R \rightarrow X$, which would lead to infinite energy dissipation according to formula (4). Could you clarify this point?

Response and Revision: Thanks for the question, we have now revised the Supplement Information to include full details of the models studied and the numerical method used (we used the Gillespie algorithm) in the Section SV entitled “Details of the Models”. In particular, for the AI model, we have calculated the energy dissipation of the reactions outside the PdP cycle (e.g., $R+X \rightarrow X$) by including reverse rates in these “outside” reactions as done in Barato and Seifert (PRE 2017) or by introducing a pool of “inactive” activators and inhibitors in the system (see the new Section SV.A in SI for details). We found that as long as the reverse rates for those reactions are fixed (which is true because they represent transcription or translation processes in real biological systems), the energy dissipation in these reactions remains constant and they can be relatively small for a reasonable choice of parameters. More importantly, the energy cost due to the reactions outside of the PdP loop remains roughly a constant independent of the key nonequilibrium parameters (γ_1, γ_2) in the PdP cycle. We have added a sentence clarifying the fact that the dissipation outside the PdP cycle in the AI model is roughly constant and does not play a direct role in regulating the oscillation of the system with the details included in the SI (section SV and Fig. S2).

6) There are examples of really bad writing in the supplement. The first paragraph of section SVI that contains equation S15 is one such example.

The authors have to improve the supplement a lot. Some parts are really hard to follow.

Response and Revision: We thank the reviewer for pointing this out. The SI is now thoroughly revised to make it more comprehensible.

Overall, there is nothing particularly new on the fact that it is possible to find models where both precision and sensitivity improve with an increase in energy dissipation. The main question is: how universal this behaviour is? In other words, under which conditions on the kinetic parameters?

Response: This is a good question. By demonstrating the results in different realistic biochemical oscillators and in the stochastic Stuart-Laudau equation analytically, we believe that one of the main results, i.e., the energy dissipation sets the bound (optimal performance limit) for both the phase diffusion and the phase sensitivity, are universal (see our response to question #4). Furthermore, the two design principles on how to choose the microscopic reaction rates to achieve these two seemingly contradictory performance limits (low phase diffusion and high phase sensitivity) at the same time are also general since they were verified in all the realistic biochemical oscillators we studied and novel (see our response to question #3). These design principles exactly answer the reviewer's question on what are the conditions under which these universal bounds can be achieved. We believe these design principles are important not only to understand the mechanisms of realistic biochemical oscillators but also to design efficient synthetic oscillators.

Point-by-Point Responses to Comments from Reviewer #2

In this manuscript, the authors proposed designing mechanisms for certain types of biological oscillators to increase both its sensitivity and robustness by varying the driven forces of free energy dissipations. This design is not allowed in the linear response regime but becomes possible in systems driven far away from equilibrium.

The work is technically interesting. The question formulated in the abstract and introduction is an important one. Their result is in terms of forward and backward fluxes which seem like nice variables to use.

However, there are two serious problems with the work that raise questions both about its validity and its relevance and thus prevent recommending this paper for publication:

Biological oscillatory systems studied here are externally driven oscillators. But the authors define phase sensitivity and phase diffusion, the two central quantities of this paper and several others, in the absence of a driving signal. Many publications have shown that the results are very different when externally driven.

The experimental connection shown in Fig 5 is a pretty far out interpretation of the experimental data. The paper would almost be stronger without it.

Without experimental data, the paper could have been an interesting theoretical exercise in computing non-equilibrium steady state properties, suitable for specialized journals. But (1) above is a serious design flaw in even metaphorically linking the analysis to the biological systems the authors mention (or indeed any imagined context). Unless the authors can derive results for oscillators with external driving - the putative subject of the paper - one cannot recommend this manuscript for publication in an interdisciplinary journal.

Response: First, we would like to thank the reviewer for the positive comments on the importance of the general problem and the soundness of our technical work. The two main criticisms turn out to be very constructive and have prompted us to carry out additional work to address them. The revised manuscript with new results on entrainment to external periodic signals incorporated and with predictions for future experiments clarified is

a much better and satisfying end product. The details of our response and changes we made are given following each specific comment below.

The above criticisms are detailed below:

1. Externally forced vs unforced systems:

As is widely appreciated, biological systems need to respond to relevant 'external cues' in environmental signals while also ignoring all kinds of irrelevant fluctuations within. The common sense way of modelling this is to define a quantity that quantifies each of these.

Here, the authors choose to study oscillatory systems like circadian clocks and the authors define a phase sensitivity χ for the former and a phase diffusion constant D for the latter concept.

However, the definition of both quantities in this paper is quite problematic. These oscillators are always used in the context of an externally time varying signal that entrains the system. The 'external cues' could then be e.g., changes in amplitude or phase of that signal. A commonly studied example would be how these oscillators overcome 'jet lag' (i.e., how they quickly respond to phase changes of the external signal).

Instead, in the authors' calculations, there apparently is no driving signal at all. It is as if the 'external cues' is the presence of an external signal in the first place.

This issue is important both for technical reasons and for motivational reasons. Technically, over the last 20 years, most papers on this subject have noted phase diffusion is very different in externally driven systems vs undriven systems - see papers by Gonze, Goldbeter etc. For more recent work on this topic, see recent work in Cell Systems from the Khammash group. Similarly, ten Wolde's group has a recent work showing that driven systems have very different phase diffusion from the undriven system modelled here.

In terms of motivation for the paper, it is also hard to imagine any situation for oscillators that operates without any entraining signal present all the time. In fact, the authors clearly recognize this since they define entrainability as a critical aspect (e.g., in the SI) and talk about

synchronizability and about Arnold tongues in different parts of the paper. And yet all the calculations of phase diffusion were done without any periodic external drive that's clearly present in all the examples mentioned in the paper and in the experimental systems this manuscript connects to.

Perhaps the authors have restricted themselves to the case of no external signal because, with an external signal, the question of dissipation becomes quite academic. E.g., the external signal may also do work on the system and it is not clear this current kind of energy budgeting is relevant. In summary, it is hard to say what real question this calculation is directly relevant for, especially at the level required for journals like this. Further, existing works show why ignoring the ever-present periodic external signal is a bad idea in defining phase diffusion.

Response and Revision: We thank the reviewer for pointing out this important issue of entrainment by external signals, which we have overlooked in the previous version of our manuscript. We have rectified this problem now by studying the entrainment process by considering the case when the internal oscillator is coupled with an external periodic signal to demonstrate the relationship between entrainability and the phase sensitivity. These new results are now included in a new subsection in the Result section entitled: "The relation between phase sensitivity and entrainment to external periodic driving" in the revised manuscript with a new figure (Fig. 4) to demonstrate the dependence of entrainment to external periodic signals on phase sensitivity and energy dissipation that we studied before.

To briefly summarize the new results, we have studied dynamics of the phase difference between the internal oscillator and the external signal by using direct simulations and the phase reduction method. We found that an initial phase difference, e.g., due to a sudden change of the phase in the external driving signal, is damped to zero with the damping rate directly proportional to the phase sensitivity studied before. As a result, the entrainment time t_e is inversely proportional to the phase sensitivity χ (see Eq. (8) and Fig. 4a in the revised manuscript) and therefore t_e decreases as more energy is dissipated as shown in Fig. 4b in the revised manuscript. Furthermore, we showed that even though the phase of the oscillator in the entrained state does not diffuse, it still fluctuates with the variance of the phase fluctuation σ^2 linearly proportional to the phase diffusion constant D we studied before in the absence of the external signal (see Eq. (9) and

Fig. 4c in the revised manuscript). Therefore, the phase fluctuation σ^2 in the entrained state also decreases with the energy dissipation as shown in Fig.4d in the revised manuscript (Note that technically it is straightforward to include the time-varying kinetic rate driven by external periodic signal in our energy dissipation calculation).

Overall, by establishing the relationship between the entrainment time and phase fluctuation with the phase sensitivity and phase diffusion that we studied before, we were able to show that the performance of entrainment characterized by the entrainment time and the phase fluctuation can be improved by free energy dissipation in the internal oscillation circuit.

2. Experimental data:

The authors rely on two kinds of possible experimental support to their theory - see Fig 5. However, neither panel shows a check of the theory.

Fig 5a: The authors first look at the temperature dependence of the phase response curve (PRC) in *Neurospora* and a couple of other organisms. The data shows that the size of the PRC generally increases with temperature but the authors want to conclude a lot more - that these relationships are linear and that their slopes are the same.

To start with, two of the organisms have only two data points! Further, the x-axis has an extremely small range (for good reason as we'll discuss later) and there are huge error bars in y, makes the slopes of these curves meaningless. To conclude that these relationships are linear and that their slopes are the same really requires a very very favorable reading of these results. Finally, a linear scaling is not enough to hang a hat on - even without any of the author's theory, many other simple assumptions would predict the same. E.g., see works on temperature compensation in clocks.

A more serious flaw is that the experiments in panel (a) were performed on living organisms. The effect of temperature on the phase response of a living organism can in no way be traced to the core oscillator's properties alone! There is a reason the x-axis of Fig 5a has such a tiny range - the organisms would die for the kind of serious temperature changes that would actually test the author's physical theory. To put another way, there is way too much biology going on in the temperature response data shown in Fig 5a to compare it to a Boltzmann factor.

For example, Ref [45], from whom the PRC data on chick pineal cells was obtained, explicitly states that the change of the PRC results from an increase in the size of the limit cycle with temperature. Such larger amplitude oscillations result from multiple biological factors as discussed in the Discussions of that paper - it is hard to imagine how one could ascribe all or even most of the effect of such a temperature change to the kind of simple Boltzmann factor analysis done here.

Similarly, Ref [46], from which PRC data on dinoflagellates were obtained, states that the PRCs due to light actually showed no temperature dependence! The authors here use the drug PRC data from the paper which does show temperature dependence, showing conclusively that the PRC's temperature dependence is not a property of the oscillator itself (the subject of the author's study) but because of the many upstream and downstream temperature-sensitive processes. Indeed, Ref [46] discusses many such explanations and tries to quantify their contributions - e.g., drugs affect protein synthesis differently at different temperatures while light does not, temperature-sensitive phase angle between overt rhythm and the pacemaker phase etc. It seems hard to conceive of a situation where all of these strong obvious temperature effects are somehow less important than the subtle temperature dependence of the oscillator itself.

I completely understand and support the author's desire to build a solid physically grounded theory of one piece of the cellular system. We certainly need more of such theory, and the above criticism should not be a mistaken for a criticism of making modelling assumptions or studying subparts of a complex system. The problem is in comparing to two data points of whole organismal experiments to such a theory, when there are clearly much stronger biological effects at play.

Finally, the putative connection to the Kai system suffers a different serious weakness. Several papers, e.g., from O'Shea lab which the authors cite, have shown experimentally that the size of the PRC is primarily set by the difference in ATP levels between day and night. The authors' theory here links the size of the PRC to the absolute ATP levels, say, in the day. But all these other publications - from the time of Arthur Winfree's book - have shown the strong impact of the difference between the day and night cycles on the height of the PRC.

Again, it is hard to imagine a situation where the change in PRCs can be attributed to the absolute ATP levels when the effect of difference in ATP levels has already been documented to be the main dominant effect. In fact, Winfree's famous classification of type 0 and type 1 PRCs is an example of how strong the latter effect is. (For general theory on this, one could consult Winfree's book or Pikowsky's book titled 'Synchronization'. For a recent experiment on the Kai system satisfying this property, see the recent Molecular Cell paper from the O'Shea lab.)

Response and Revision: We thank the reviewer for these insightful comments regarding possible experimental evidence/verification for our theoretical work. There are two comments here – one for the *in vivo* systems we found in the existing literature on PRC in different temperature as shown in the original Fig. 5a and the other for the Kai system in cyanobacteria. We address these two comments separately in the following.

(1) The measured PRC at different temperatures in different *in vivo* systems: The reason we mentioned these experiments is to search for existing experiments that measure the PRC amplitude to a given perturbation (light, temperature, drug) under different energetic conditions, in particular at different background ATP/ADP ratios. Our theory predicts that the PRC amplitude for a given perturbation (e.g., a rise in temperature from 30C to 35C in a 0.1 period window) will decrease as the background ATP/ADP ratio decreases. However, we did not find such experiments in existing literature. The closest thing we found was the temperature-dependent PRC curves that we included in the manuscript. Interestingly, in the experimental systems we found, Neurospora, Gonyaulax polyedra and chick pineal cell, except for the case of light response in chick pineal cell where no obvious T - dependence is found, PRC amplitudes indeed increase with T_0/T , which seems to be consistent with our theory. However, we agree with the reviewer that we need to be very careful about interpreting these temperature dependent PRC experiments due to the fact that many kinetic rates may depend on temperature.

Revision: We have rewritten/shortened and significantly toned down the discussion on the temperature-dependent PRC measurements in the existing literature. We now present them only as the start of our efforts to find experimental test of our theory and caution the readers about

interpretation of the data by stating explicitly in the revised manuscript that: “we need to caution that many internal variables in biological systems may depend on T, therefore these temperature-dependence measurements in *in vivo* systems may not serve as direct tests of our theory.” We have also moved Fig. 5a to the SI (Fig. S11) to avoid any undue attention to these data beyond what we described in the revised manuscript. We believe the revised discussion on these existing data is appropriate and it set the stage for the next paragraph where we propose explicitly a doable experiment in the Kai system that can be used directly to test our theory.

(2)The Kai system: The best place to test our theory is the *in vitro* Kai system. We think our work presented in this manuscript and the corresponding predictions are new, beyond what has already been done/known in the literature for the Kai system. In particular, for the data published in the 2011 Science paper by Rust-Golden-O’Shea as mentioned by the reviewer, the background ATP is kept at nearly 100% (ADP level is 0% to start with), the perturbation they used to obtain the PRC is the drop in ATP level or the increase in the ADP level (Fig. 2E in the Rust paper) during a small pulse of time window (~4hrs). When the strength of the perturbation, i.e., the level of ADP% during the pulse (fixed at 4hrs), is increased, the PRC amplitude increases linearly with the strength of the perturbation strength. This is simply due to linear response theory, totally consistent with any reasonable models including ours. However, this is not the main point of our work. The main prediction of our work is that if the ATP% in the background changes (e.g., from 99% ATP to 60% ATP), then the PRC for a fixed perturbation (both strength and duration) will change with the background ATP%. In particular, the PRC amplitude will be larger in a background with the higher ATP% (99%) than that in a background with the lower ATP% (60%).

Revision: We have rewritten this part of the discussion and introduced a new figure (Fig. 6) to propose a clean and doable experiment to test our theory in the *in vitro* Kai system. In the 2011 Rust paper, the ATP% is used as the perturbation itself with the background ATP% kept close to 100%. To test our theory, we propose to use temperature as the perturbation for the PRC measurements. Our predictions can be tested by measuring the temperature-PRC at different background ATP% levels. Changing

background ATP% is achievable as demonstrated in the Rust et al 2011 Science paper and subsequent work from the Rust group (e.g., C. Phong et al, PNAS 2013). To make our predictions more quantitative, we have now calibrated the AI model, to which the Kai system belongs, to generate the same temperature-PRC shape as the one measured by Yoshida et al (Yoshida et al, PNAS 106, 1648-1653 (2009), see the new Fig. 6a for details. Then, the PRC for the same perturbation but for reduced levels of energy dissipation can be computed. As shown in the new Fig. 6b, the PRC amplitude decreases with lowered level of energy dissipation. These temperature-PRC's at different energy dissipation levels (or different background ATP/ADP ratios) serve as explicit predictions for future experiments to test. We have been in contact with Michael Rust, who believe these proposed experiments haven't been done before and they can be carried out to test our theory.

Point-by-Point Responses to Comments from Reviewer #3

Fei et al. present a detailed analysis of how energy dissipation in oscillating reaction network is related to phase sensitivity and phase diffusion behavior. The work follows up on a previous work by authors which studied phase diffusion in oscillatory biochemical reactions. The current work mainly focuses on the relation between energy dissipation and phase sensitivity. The relation is illustrated on several simple models, but a general relation between the amount of dissipated energy and phase sensitivity is derived for all oscillatory circuits. I consider the work to be innovative and important, the results are sound and make testable predictions. In the present form, I would find the work suitable for PRX or similar journal, but for the broad audience of Nature Communications, I would ask for expansion of the part of the article that makes explicit connections with known biological oscillators.

In particular, it is not clear to me that phase sensitivity is a general important feature for biological circuit. For instance, the circadian oscillations are synchronized by a long term exposure to daylight, rather than by a short perturbation. While the author's results on the relation between dissipated energy and phase sensitivity holds in general, it might not be the case that circuits are designed for achieving best phase sensitivity for a given amount of dissipated energy. In section C, predictions are made for the circuit properties that would optimize phase sensitivity / phase diffusion for a given amount of energy W . Can a comparison be made to parameters of known biochemical oscillator, such as the examples used in Fig 5a?

Response: There are two main questions raised by the reviewer. The first main question is about the relevance of phase sensitivity for entrainment by periodic external signals, which is the same question asked by reviewer #2 (see our response to reviewer #2 regarding this question). We agree that it is important to understand the relationship between phase sensitivity and entrainment (or synchronization) by external periodic signals. We have now included a new subsection entitled “The relation between phase sensitivity and entrainment to external periodic signals” in the Results section. To summarize the new results briefly, we showed by both theoretical analysis and numerical simulations that the entrainment time t_e , i.e., the time it takes to synchronize with an external signal after a phase perturbation (e.g., jet-

lag), is inversely proportional to the phase sensitivity (see Eq. (8) and the new Fig. 4a in the revised manuscript). This means that the higher the phase sensitivity the shorter it takes to entrain with the external signal, which is a highly desirable property of the clock. We also showed that both the entrainment time and the phase fluctuations in the entrained state decrease with increased free energy dissipation (see Fig. 4b&c&d). Given the importance of entrainability for biological clocks (we all have the desire to get rid of the jet-lag as soon as possible), we believe it is reasonable to assume that biological clocks have evolved to enhance phase sensitivity and reduce phase fluctuation by using some of their resources including metabolic energy. As we showed in our work, energy dissipation is only a *necessary condition* to enhance phase sensitivity and maintain low fluctuation at the same time, there are infinite ways of wasting energy without achieving the desired effects. One main focus of our work is to discover the design principles to achieve the desired effects (high phase sensitivity and low phase fluctuation) with a fixed energy budget.

The second main question is about direct comparison of our results with existing experiments. This is a very good question. However, for most of the experimental clock systems the biochemical rate constants in the underlying biochemical networks have not been measured, and in many cases the key biomolecules are not all identified. The best characterized clock system is the Kai system in cyanobacteria where the key players (KaiA, KaiB, and KaiC, are identified). We have now analyzed a specific model of the Kai system with parameters measured by *in vitro* experiments to compare with our theory. A careful study of the experimentally measured parameters in the Kai system showed strong evidence in support of the two general design principles found in our study.

Revision: We have now revised the manuscript to address the two main question raised by the reviewer: (1) We have now added a new subsection entitled “The relation between phase sensitivity and entrainment to external periodic signals” on pages 3-4 and a new figure (Fig. 4) in the Result section to describe the relation between entrainment to external periodic signals and the phase sensitivity and phase diffusion we studied before; (2) we have now added the new experimental evidences in support of the design principles in the kai system entitled “Experimental evidence in the Kai system” on page 6 right after we describe the two design principles in in the Result section.

Further points:

1) The AI model is one of the main models used to illustrate the theoretical results, and should be hence presented in main text too (Figure S1). Various combinations of values of parameters γ_1 and γ_2 are studied, I would be curious to include in the plot the typical values of $\gamma = \gamma_1 \gamma_2$ (ie plugging into the gamma formula the typical concentrations of ATP, ADP, P and K_{eq}) encountered in PdP cycle in living systems.

Response and Revision: Due to space limitation in the main text, we could not include the AI model and its results in the main text. They are included in the SI. If the reviewer feels strongly about it, we could combine Fig. S1 with Fig. 2 in the main text. However, that may take away space in Fig. 2 to describe the existence of multiple pathways between the chemical states (“X” and “Y”), which is crucial in deriving the design principles in our study. In addition, the AI model was shown/illustrated in our previous paper (see Fig. 1b in Cao et al, Nat. Phys. 2015), so we do not feel it is absolutely necessary to show it again here.

About the reviewer’s second question, we agree that It is a good idea to include a typical value of γ in the performance plot for the AI model with the PdP cycle. In a living organism, the ATP hydrolysis energy is roughly $12k_B T$, which makes $\gamma = \gamma_1 \gamma_2 \approx e^{-12} \approx 10^{-5.2}$. We have now included this typical value and show it as lines in the revised Fig. S5 in the SI.

2) In the Brusselator model, why is k_{-1} and k_{-1} not included in the forward/ backward rates?

Response and Revision: We are not sure exactly what the reviewer’s question is. In our calculations of both the dynamics and the energetics of the system, we have included both the k_1 and k_{-1} rates. Perhaps the question is in the context of the design principles? In that regard, the focus of our work here is to study the design principles on how to choose the kinetic rates within the key nonequilibrium reaction cycle (e.g., the cycle shown in Fig. 2), which drives the oscillation and determines its performance. Therefore, the forward and backward fluxes we considered in this work for the design principles are those within the irreversible reaction cycle.

3) The biological oscillators studied in Fig 5 should be briefly introduced and described in Supplementary Information

Response and Revision: In the revised manuscript, we choose to focus on a specific biological oscillator, the cyanobacteria circadian clock (the kai system), as all the relevant components of the system are known and there are available quantitative measurements of the underlying kinetic rates (see our response to one of the reviewer's main questions). The details of the kai system and a simple kinetic model (section SXI) with all the measured parameters (Table S1) are given in the SI.

4) In the title of the article, I would suggest replacing "biochemical networks" by "biochemical oscillatory systems", as this more accurately describes the studied system

Response and Revision: We have changed the title according to the suggestion of the reviewer.

5) Fig 5b and S9 captions should include also include description of blue data points in the inset

Response and Revision: We have added descriptions in these figures to describe all data points shown.

REVIEWERS' COMMENTS:

Reviewer #1 (Remarks to the Author):

I have read the answer to my questions carefully. I also read the manuscript. I did not check the supplement: I am trusting the authors improved it. Overall, I am quite satisfied with the answers to my questions. I think this manuscript is a nice contribution and of interest for people working on biophysics and statistical mechanics/non-equilibrium thermodynamics. I do have some further observations related to the answers but my recommendation is that the manuscript should be accepted. It is original and will generate further work on the relation between thermodynamics and biophysics.

Concerning question 2

I am not sure the authors have given a general definition of phase diffusion. I do know how to define τ_c (as in Qian's paper) for any model but phase diffusion seems to be a tricky thing. It seems to me that their claim in the previous paper is that the relation between phase diffusion and τ_c is a result and not a definition of phase diffusion. I would also know how to calculate it on a simulation of one of their models but I would have trouble defining it precisely. I guess it is related to some first passage time quantity. The authors might want to further discuss this point: can you present an equation

with a precise definition of phase diffusion?

Concerning point 4

As the authors say, their result is a limit on optimal performance. As such they might be able to express it as an inequality. I am happy enough with their presentation, stating the result as an equality and stating it is relation valid for the optimal performance. However, if they could express it as an universal inequality it would be better.

Even though the results from the PRE paper I mentioned are largely inspired by their nat phys paper, there is a big difference between both results. The result from the PRE paper is expressed as an inequality that is true for any stochastic process. Furthermore, it does not depend on the rate of energy dissipation but rather on the thermodynamic force γ (i.e. the inequality is independent of kinetic parameters). γ is something one would typically know ($\gamma=20$ for ATP in physiological conditions). The authors might want to further comment about that or maybe even try to express their results as inequalities (if it is possible).

Reviewer #2 (Remarks to the Author):

I have gone through the authors' response and the revised paper. I find it hard to recommend publication because of two reasons:

a. the theoretical physics contribution here does not go beyond several statistical physics papers published in recent years,

b. the biological contribution in connecting to realistic clocks is weak at best and often incorrect or mischaracterizes prior experimental work.

a. many statistical physics works in recent years have show the basic idea of this paper. The main result of this paper is not news to anyone in the field. However, the earlier papers (e.g., many by Barato/Seifert) use simplified models to get at the heart of the question. The authors here seem to want to extend those ideas to more realistic models.

However, the complications introduced here don't make things more realistic - they don't account for any major feature of real clocks. For example, the authors still do not really account for driving forces. In this revision, the authors have added a section on how weak driving gives variances that depend on various quantities. But their trade-off and analysis is still in terms of the undriven quantities and not the full driven quantity. Thus, the paper does not account for the most obvious feature of biological clocks - they are strongly driven by the external environment!

So I don't see how this paper is an advance over the numerous abstract statistical physics papers on the same topic.

b. The above lack of significant new theory would be OK if this paper connected those earlier ideas to experiments. But the paper fails here for a different reason.

I appreciate the authors removing some of the misleading experimental figures and related statements from the previous version. The cited experimental paper attributed the PRC effect to possible degradation of drugs at different temperatures - a far cry from the second law of thermodynamics!

However, the revised section on Kai clocks is also poorly done. The new Fig 6 has a caption that when casually read, sounds like a great experimental link. However, the results shown are just about

sensitivity ξ and reversibility γ - this relationship does not test the ideas about energy, diffusion and sensitivity laid out in the paper.

Further, in the main text, the authors focus on one aspect of O'Shea's 2011 paper while completely ignoring all subsequent and prior work. All these other works suggest that the authors' prediction based on 'reference' value of a physiological parameter like ATP is completely swamped by a much bigger effect - namely, the change in the physiological parameter. As a recent example, in Aug 2017, O'Shea's lab had a publication in Molecular Cell where concepts relevant to current paper were tested. Numerous other works on Drosophila and other clocks show the same result - e.g., see the book on insect clocks by Saunders or any recent review of Winfree's limit cycle work. In all of these works, the size of the PRC is mostly set by the change in physiological parameters like ATP and temperature and not the 'reference' value of that parameter. (Even the concept of a 'reference' value is just not used in these experimental papers, for good reason.)

To summarize, I don't see this paper going beyond prior work either as a work of theoretical physics or in building a bridge to biology. I don't see how this rises to the standards of novelty / impact asked for by journals like Nat. Comm.

Reviewer #3 (Remarks to the Author):

I am pleased by the modifications that the authors made to the manuscript that satisfy the concerns I raised previously. I think especially the comparison to Kai system and list of testable predictions section make this now a better paper, suitable for publication in Nature Communications.

Responses to comments from Reviewer #1

Concerning question 2

I am not sure the authors have given a general definition of phase diffusion. I do know how to define τ_c (as in Qian's paper) for any model but phase diffusion seems to be a tricky thing. It seems to me that their claim in the previous paper is that the relation between phase diffusion and τ_c is a result and not a definition of phase diffusion. I would also know how to calculate it on a simulation of one of their models but I would have trouble defining it precisely. I guess it is related to some first passage time quantity. The authors might want to further discuss this point: can you present an equation with a precise definition of phase diffusion?

Response: Phase diffusion is defined following the general way of characterizing diffusive processes. From the dependence of the phase variance on time, i.e., $\sigma_\varphi^2(t) \equiv \langle (\varphi(t) - \langle \varphi(t) \rangle)^2 \rangle = D_\varphi t$, we can define the phase diffusion constant D_φ . In Supplementary Eq. 10, we provide a continuous phase equation which could be formally deduced from the original chemical Langevin equation (CLE) by using the phase reduction method. From this phase equation with noise (Eq. 10 in the SI), the phase variance can be computed and as we showed in Supplementary Eq. 13, the phase behaves in a diffusive with a phase diffusion constant that depends on the phase noise strength.

We would like to point out that the relation between phase diffusion constant and autocorrelation time τ_c is not obtained by definition, it is a derived result as shown in Supplementary Eq. 17. In practice, it's very difficult to compute the phase diffusion constant by deriving the phase equation from CLE. Since we only need the asymptotic diffusion constant (but not the detailed information in a short time less than one period), therefore we calculate the correlation function of a key state variable from our simulations to obtain the correlation τ_c , from which we can then determine the diffusion constant.

Revision: We have added the general definition of phase diffusion in the paragraph above Eq. (2). To clarify the relation between phase diffusion and correlation time, we have now revised our manuscript (right before Eq.

(2)) to explicitly state that the finite correlation time of stochastic oscillation is due to, but not a definition of, the diffusive phase dynamics. And this relationship provides a more practical way of computing the phase diffusion constant.

Concerning point 4

As the authors say, their result is a limit on optimal performance. As such they might be able to express it as an inequality. I am happy enough with their presentation, stating the result as an equality and stating it is relation valid for the optimal performance. However, if they could express it as an universal inequality it would be better.

Even though the results from the PRE paper I mentioned are largely inspired by their nat phys paper, there is a big difference between both results. The result from the PRE paper is expressed as an inequality that is true for any stochastic process. Furthermore, it does not depend on the rate of energy dissipation but rather on the thermodynamic force γ (i.e. the inequality is independent of kinetic parameters). γ is something one would typically know ($\gamma=20$ for ATP in physiological conditions). The authors might want to further comment about that or maybe even try to express their results as inequalities (if it is possible).

Response: This is an excellent suggestion. Actually, in the models we studied, there exist a minimal diffusion constant (D_{\min}) and a maximal sensitivity (χ_{\max}) for a given energy dissipation ΔW per period as shown by the dashed lines in Figure 3 and Supplementary Figure 4&5. This best performance of phase fluctuation is consistent with the inequality in paper of Barato and Seifert PRL 2015. We have modified our text to explicitly introduce these optimal performance limits and express our results in the form of an inequality for the model parameters we have varied.

We would like to point out that both the optimal sensitivity and the minimum phase diffusivity depends on ΔW , the energy dissipation per period (in unit of kT), which means that they don't depend on specific kinetic constants. This makes it suitable to compare ΔW directly with the number of ATP hydrolyzed per period.

Revision: We have now included the minimal diffusion constant (D_{\min}) and the maximum sensitivity (χ_{\max}) in the text and in Fig. 3, re-expressed our Eq. 6 as an inequality to clarify the best performance, and also added another citation to a previous work (Barato and Seifert PRL 2015).

Responses to comments from Reviewer #2

Reviewer #2 (Remarks to the Author):

I have gone through the authors' response and the revised paper. I find it hard to recommend publication because of two reasons:

- a. the theoretical physics contribution here does not go beyond several statistical physics papers published in recent years,
- b. the biological contribution in connecting to realistic clocks is weak at best and often incorrect or mischaracterizes prior experimental work.

a. many statistical physics works in recent years have show the basic idea of this paper. The main result of this paper is not news to anyone in the field. However, the earlier papers (e.g., many by Barato/Seifert) use simplified models to get at the heart of the question. The authors here seem to want to extend those ideas to more realistic models.

However, the complications introduced here don't make things more realistic - they don't account for any major feature of real clocks. For example, the authors still do not really account for driving forces. In this revision, the authors have added a section on how weak driving gives variances that depend on various quantities. But their trade-off and analysis is still in terms of the undriven quantities and not the full driven quantity. Thus, the paper does not account for the most obvious feature of biological clocks - they are strongly driven by the external environment!

So I don't see how this paper is an advance over the numerous abstract statistical physics papers on the same topic.

Response: Indeed, employing tools from non-equilibrium statistical physics to understand biological functions has been an active area in the community recently. For oscillatory systems, in particular, most of them focus on the role of entropy production suppressing phase diffusion/period fluctuations (e.g., in Cao et al 2015 Nat. Phys. , and Barato and Seifert 2017 PRE). However, here we mainly focus on how free energy dissipation can be utilized to enhance phase sensitivity. By studying biochemical network models, we also derived novel design principles for increasing

precision and sensitivity, which is different from the ones studied in the simplified model in Barato and Seifert (PRE 2017). The key difference is that multiple reaction pathways, rather than just one single chemical process, connecting nearby microscopic states are considered explicitly in our analysis.

In the revised manuscript, we do account for the periodic external driving force having the same (or nearly the same) period of the intrinsic limit cycle oscillation. As a result, the oscillation phase of the system can now synchronize to and fluctuate around the environmental reference. Although we only did linear perturbation analysis for small input, entrainment is essentially a weak driven process (as considered in Monti et al 2017 arXiv paper and Gupta et al 2016 Cell Systems paper). If the coupling is strong enough, any system can be driven to oscillate by periodic forcing. However, the internal system has to oscillate by itself to be entrained by weak forcing, and most biochemical systems do have their own intrinsic autonomous oscillation. It is thus clear that entrainment behavior must be dependent on both the external driving strength and internal property of the system. As shown in Eq. 8&9, besides being linearly proportional to the signal strength ε , both the time needed for synchronization t_e and the phase variance σ around the phase of the external driving signal are also related to the system's internal properties, namely the phase sensitivity χ and diffusion constant D .

The general discussion provided in Section C. "The relation between phase sensitivity and entrainment to external periodic driving" can be applied to specific models (such as Stuart-Landau equation) without difficulty. The key idea is that the entraining process can be and should be connected to the oscillators' internal properties.

Revision: We have now added a statement in the revised manuscript (in between Eq. (8) and Eq. (9)) to clarify the point that the external driving and the intrinsic property of the system together shape the entrainment process.

b. The above lack of significant new theory would be OK if this paper connected those earlier ideas to experiments. But the paper fails here for a different reason.

I appreciate the authors removing some of the misleading experimental figures and related statements from the previous version. The cited experimental paper attributed the PRC effect to possible degradation of drugs at different temperatures - a far cry from the second law of thermodynamics!

However, the revised section on Kai clocks is also poorly done. The new Fig 6 has a caption that when casually read, sounds like a great experimental link. However, the results shown are just about sensitivity χ and reversibility γ - this relationship does not test the ideas about energy, diffusion and sensitivity laid out in the paper.

Further, in the main text, the authors focus on one aspect of O'Shea's 2011 paper while completely ignoring all subsequent and prior work. All these other works suggest that the authors' prediction based on 'reference' value of a physiological parameter like ATP is completely swamped by a much bigger effect - namely, the change in the physiological parameter. As a recent example, in Aug 2017, O'Shea's lab had a publication in Molecular Cell where concepts relevant to current paper were tested. Numerous other works on Drosophila and other clocks show the same result - e.g., see the book on insect clocks by Saunders or any recent review of Winfree's limit cycle work. In all of these works, the size of the PRC is mostly set by the change in physiological parameters like ATP and temperature and not the 'reference' value of that parameter. (Even the concept of a 'reference' value is just not used in these experimental papers, for good reason.)

Response: In principle, the most direct test of our theory would be some experiments measuring the relation between phase sensitivity χ and dissipation ΔW in real systems, as we did in Figure 3. One can quantitatively measure ΔW by, for example, measuring the ATP hydrolysis rate. However, for current experimental techniques, we think it's much easier (and cleaner) to change the ratio of ATP and ADP concentration (or equivalently γ), which can be done experimentally in vitro. The free energy dissipation increases with the ATP/ADP ratio. (Supplementary Figure 2).

As for the recent experimental works in the Kai system, we are fully aware of them, for example the recent paper by Gan and O'shea (Mol. Cell 2017) mentioned by the reviewer. However, we did not find any solid evidence either supporting or contradicting our theory. For example, in Gan & O'Shea paper, the authors used temperature pulse to explore the

geometric structure of the system's limit cycle. By systematically scanning the pulse duration Δt and the initial circadian phase ϕ_{ini} at the time the pulse was given, they found multiple critical perturbations that induce stochastic phases following a 25°C pulse and thereby cause attenuation of circadian oscillations at the population-averaged level. This phenomenon is well understood theoretically as “black holes” of phase resetting, where the perturbed limit cycle cuts through the unstable singularity of the original “reference” limit cycle (see for example Murray's “Mathematical Biology” for related). Gan and O'Shea did further experiments to verify this limit cycle framework for interpreting the indeterminate phases under critical condition. Our theory is consistent with the Gan & O'Shea work in general in at least the following two aspects: (1) a limit cycle model is sufficient for describing the induced phase shift; and (2) the geometric structure of the initial “reference” limit cycle is crucial to the oscillator's phase response property. However, for the following two reasons, we find it hard to connect Gan & O'Shea (Mol. Cell 2017) with our theory directly (not to mention negating the claims made in our study):

- (1) Our theory predicts that given the same amount of perturbation, phase shift would change with different “reference” energetic state of the system (characterized by γ or ΔW). By contrast, their experiments fixed the “reference” state (starting from the same background temperature) and varied the perturbation strength variable Δt .
- (2) Our theory is derived from linear perturbation analysis under the assumption that the perturbation is small, yet the interest of the experiments, i.e. “black holes”, happens only for sufficiently large perturbation $\Delta r / r \sim 100\%$ of the limit cycle. Such critical condition clearly goes far beyond the assumption of our theory.

Despite lacking direct relevance to our theory, Gan & O'Shea's work clearly demonstrates a powerful experimental method for exploring the geometric structure of the isochrons in the system, which we studied theoretically using the phase reduction method. In fact, the quantitative techniques available and the relative simplicity of the Kai system are exactly the motivations for us to place our prediction in the Kai system.

We agree that the change in the physiological parameter, which serves as the phase resetting signal, would also affect the response $\Delta\phi$. In Supplementary Equation 5, we identified two factors, besides the perturbation strength, that determine the phase shift — one perturbation-

independent parameter χ that characterizes the limit cycle geometry and depends solely on the “reference” state, and another perturbation-dependent factor $dF/d\mu$. Throughout our paper we have assumed that the latter factor does not vary much for different “reference” states, i.e. the same perturbation applied to the system would cause similar $\Delta r / r$ to the limit cycle geometry, which should be the case when the system is away from the onset of oscillation. Therefore, under this assumption, the response sensitivity χ should dominate the phase response. However, this assumption cannot be proved rigorously, therefore, we turned to direct numerical simulations. The results as we presented in Fig. 6 seem to be consistent with this assumption. Based on these considerations, we believe the predictions made in our paper about the Kai system as shown in Fig. 6 represent doable and realistic tests of our theory. We have discussed with an expert experimentalist in the Kai system, Prof. Michael Rust from University of Chicago, about this particular prediction. Michael Rust believes it is an interesting and doable experiment.

Revision: We have now explicitly stated our assumption right after introducing the concept of phase sensitivity in Eq.(1). To make it clearer, we further reiterate the hypothesis we made about the relative amplitude change before we present our predictions for experimental tests in the last paragraph of the paper.

To summarize, I don't see this paper going beyond prior work either as a work of theoretical physics or in building a bridge to biology. I don't see how this rises to the standards of novelty / impact asked for by journals like Nat. Comm.

Reviewer #3 (Remarks to the Author):

I am pleased by the modifications that the authors made to the manuscript that satisfy the concerns I raised previously. I think especially the comparison to Kai system and list of testable predictions section make this now a better paper, suitable for publication in Nature Communications.